# Synthetic Cathinones and Neurotoxicity Risks: A Systematic Review

**DOI:** 10.3390/ijms24076230

**Published:** 2023-03-25

**Authors:** Gloria Daziani, Alfredo Fabrizio Lo Faro, Vincenzo Montana, Gaia Goteri, Mauro Pesaresi, Giulia Bambagiotti, Eva Montanari, Raffaele Giorgetti, Angelo Montana

**Affiliations:** 1Department of Excellence Biomedical Sciences and Public Health, Marche Polytechnic University, 60121 Ancona, Italy; gloriadazi@gmail.com (G.D.); fabriziolofaro09@gmail.com (A.F.L.F.); g.goteri@univpm.it (G.G.); m.pesaresi@univpm.it (M.P.); giuliabamba@gmail.com (G.B.); eva.montanari@ospedaliriuniti.marche.it (E.M.); r.giorgetti@univpm.it (R.G.); 2Dipartimento di Anestesia, Rianimazione e Emergenza-Urgenza, Fondazione IRCCS (Istituto di Ricovero e Cura a Carattere Scientifico) Ca’ Granda Ospedale Maggiore Policlinico, 20122 Milan, Italy; montana_vincenzo@libero.it

**Keywords:** new psychoactive substances, synthetic cathinones, neurotoxicity, intoxication, fatality, animal and human studies, brain damage

## Abstract

According to the EU Early Warning System (EWS), synthetic cathinones (SCs) are the second largest new psychoactive substances (NPS) class, with 162 synthetic cathinones monitored by the EU EWS. They have a similar structure to cathinone, principally found in Catha Edulis; they have a phenethylamine related structure but also exhibit amphetamine-like stimulant effects. Illegal laboratories regularly develop new substances and place them on the market. For this reason, during the last decade this class of substances has presented a great challenge for public health and forensic toxicologists. Acting on different systems and with various mechanisms of action, the spectrum of side effects caused by the intake of these drugs of abuse is very broad. To date, most studies have focused on the substances’ cardiac effects, and very few on their associated neurotoxicity. Specifically, synthetic cathinones appear to be involved in different neurological events, including increased alertness, mild agitation, severe psychosis, hyperthermia and death. A systematic literature search in PubMed and Scopus databases according to PRISMA guidelines was performed. A total of 515 studies published from 2005 to 2022 (350 articles from PubMed and 165 from Scopus) were initially screened for eligibility. The papers excluded, according to the criteria described in the Method Section (n = 401) and after full text analyses (n = 82), were 483 in total. The remaining 76 were included in the present review, as they met fully the inclusion criteria. The present work provides a comprehensive review on neurotoxic mechanisms of synthetic cathinones highlighting intoxication cases and fatalities in humans, as well as the toxic effects on animals (in particular rats, mice and zebrafish larvae). The reviewed studies showed brain-related adverse effects, including encephalopathy, coma and convulsions, and sympathomimetic and hallucinogenic toxidromes, together with the risk of developing excited/agitated delirium syndrome and serotonin syndrome.

## 1. Introduction

The illicit market and related misuse of drugs has expanded to cover alternative substances [1], such as New Psychoactive Substances (NPSs). NPSs comprise a heterogenous group of substances [2], such as prescription drugs and research chemicals, which are not controlled under the 1961 Single Convention on Narcotic Drugs or the 1971 Convention on Psychotropic Substances, and are synthetized to mimic the psychoactive effects of common drugs of abuse. Consumption of these drugs of abuse [3,4,5,6,7,8,9,10] can cause several adverse effects such as acute psychosis, bradypnea, angina pectoris, migraine, headaches, and life-threatening cardiovascular problems that range from mild tachycardia to arrhythmias, myocardial infarction and even death [11,12,13].

At the end of 2021, the European Monitoring Centre for Drugs and Drug Addiction (EMCDDA) was monitoring around 880 new psychoactive substances, and in 2020 alone, cathinone powders represented 65% of the materials seized. A quarter of this was 3-methylmethcathinone (3-MMC) and 3-chloromethcathinone (3-CMC), while N-ethylexedrone represented a third of such seizures [2].

Synthetic cathinones (SCs) are also known as “Bath salts”; they are “legal” replacements of controlled stimulants derived from Catha Edulis (khat plant). The khat plant is indigenous to eastern Africa and the Arabian Peninsula. SCs are identified as designer drugs because their chemical structures are easily modified in order to circumvent legal controls and they are designed to mimic the effects of more traditional psychostimulants. They are derived from phenylalkylamines and are analogues of beta-ketone amphetamine (AMPH); indeed, they act in a similar way to AMPH by releasing psychostimulants [2,14,15,16,17].

Some of the common neurological effects due to the intake of SCs are euphoria, increased drive, loquacity, psychomotor agitation, hallucinations, increased alertness, empathy, energy, paranoia, delirium [14,18,19,20].

Mechanisms of action for synthetic cathinones involve interactions with dopamine, serotonin and norepinephrine transporters with varying affinities and selectivities. This has been shown in in vitro studies, including human cell lines and preclinical models of ring substituted cathinones, such as methylone, which act as transporter substrates that increase the release of dopamine, serotonin, and norepinephrine. Substances possessing a pyrrolidine ring, as in α-PVP (α-pyrrolidinopentiophenone), act as transport blockers (reuptake inhibitors) at the dopamine transporter. Increasing the length of the α-carbon chain increases the affinity and potency at the dopamine transporter. Compounds with a higher potency at the dopamine transporter, includincludeding α-pyrrolidinophenones and 4-fluoroamphetamine (4-FA), exhibit stimulant properties similar to methamphetamine while cathinones that have similar potencies at dopamine and serotonin transporters, or higher potency at the serotonin transporter, may have more empathogenic activity (e.g., ethylone) [11,17,21]. Some of the direct neurotoxicity effects are hyperthermia and neuroinflammation [21]. “Bath salts” are usually sold as brown or white crystal-like powder [22] and are consumed through insufflation, oral or intravenous routes [23]. They are often labelled as “legal highs”, “plant food”, “not for human consumption”, “research chemicals” and are not expensive [21]. Synthetic cathinones that have been found in these products include butylone, dimethylcathinone, ethcathinone, ethylone, 3- and 4-fluoromethcathinone (3-FMC, 4-FMC), mephedrone, methedrone, methylenedioxypyrovalerone (MDPV), methylone, and pyrovalerone (see Figure 1) [24,25,26]. After consumption of SCs, the desired effects appear in 30–45 min and last from 1 to 3 h [27].

Different studies have confirmed the ability of SCs to stimulate the central nervous system (CNS) with an increase in the subject’s need to move [14], associated in some cases with aggressive, uncontrollable behavior or loss of consciousness [22].

The aim of this review is to identify the neurotoxic effects caused by exposure to synthetic cathinones, by evaluating studies conducted on animal models and with humans.

## 2. Materials and Methods

A systematic literature search was performed in PubMed, Scopus, Web of Science databases and official international organizations’ websites, according to PRISMA guidelines, Figure 2 [28]. Keywords were “new psychoactive substances”, “synthetic cathinones”, “bath salts”, “central nervous system”, “neurotoxicity”, “effects”, “brain”, “rat”, “autopsy”, “intoxication”, “human” and “blood-brain barrier”, “fatalities” and they were searched in different combinations by two scientists individually.

A total of 515 scientific articles (350 from PubMed and 165 from Scopus) published from 2005 to 2022 were initially screened for eligibility. Two scientists individually evaluated each entry recovered from a database; titles and abstracts mentioning SC studies and neurotoxic effects on humans and rats were further considered for full-text reading.

The screening process excluded studies according to the following criteria:(1)Articles not written in English;(2)Commentary, editorial letters, surveys;(3)Removal of duplicates;(4)Irrelevant studies.

Finally, articles were excluded for insufficient data.

## 3. Results and Discussion

A total of 515 studies were initially screened. Of these, 330 articles from PubMed and Scopus were excluded, according to the criteria described in the Methods section. The remaining 240 records were further assessed for eligibility and articles without sufficient data were excluded. Finally, 76 studies were analyzed and classified as “finding on animal models” (n = 56) or “finding on humans” (n = 20) according to the samples investigated. All results are summarized in Table 1, Table 2 and Table 3. Figure 3 summarizes current understanding of SCs mechanism of action and effects on rat and human brain.

### 3.1. Findings on Animal Models In Vitro and In Vivo

Different types of toxicity can occur following exposure to SCs, notably, neurotoxicity, cardiotoxicity, nephrotoxicity and pulmonary toxicity. In this section, we focus attention on the neurotoxic potential of SCs, since the main mechanism of action of these substances is dysregulation of the monoamine systems. Indeed, the neuro-clinical manifestations (e.g., toxidromes) reported in SC-induced human intoxications tally with the monoamine dysfunction observed in animal studies [29].

The results are summarized in Table 1.

Regarding the studies carried out on animal models, 56 articles that employed in vivo or in vitro experiments on cell cultures were considered (Table 1). The most studied SCs were mephedrone, methylone and α-pyrrolidinopenthiophenone (α-PVP). The studies evaluated effects on the locomotor system, the ability of SCs to cross the blood-brain barrier (BBB), and neurotoxicity at different targets and interaction pathways.

To date, researchers have conducted various types of experiment to understand how SCs interact with the central nervous system, especially their effects on locomotory activity.

Studies on animal models, in particular on rats, show that treatment with SCs, such as mephedrone, cause ambulatory hyperactivity. Shortall et al. described a sensitization to intermittent treatment with intraperitoneal injections (i.p.) of mephedrone (1, 4 or 10 mg/kg) and they noticed a significant ambulatory hyperactivity after the sixth dose [30]. This is due to dopaminergic neuronal activity and dopamine release in the nucleus accumbens of the rats.

Several studies demonstrated that ‘second-generation’ mephedrone analogs such as 4-methyl-N-ethylcathinone administrated to rats through i.p. or intravenous (i.v.) injections caused a stimulation of locomotor activity [31,32].

This result was confirmed by other research, including the study by Javadi-Peydaret al., which showed that intravenous administration of pentylone, pentedrone and methylone in rats increased locomotor activity as a result of their psychostimulant effects. In particular, the stimulating effects caused by the intake of pentedrone lasted longer than those caused by the other two substances (at the same dose); the effects were equivalent for male and female rodents [33]. This stimulant capability of pentylone was confirmed by Saha et al. who showed that the pentylone locomotor stimulant effect was higher than the butylone effect in mice [34]. Moreover, pentylone and eutylone were stronger locomotor stimulants than butylone in treated mice [35]. Increase of locomotor activity in mice after exposure to SCs seems to be common; indeed, other studies have demonstrated that exposure to 3-chloromethcathinone, 4-chloromethcathinone, 4-fluoro-α-pyrrolidinopentiophenone and 4-methoxy-α-pyrrolidinopentiophenone also produced an increase of horizontal spontaneous locomotor activity with a dose-dependent effect, while α-pyrrolidinophenones increased vertical spontaneous locomotor activity. Furthermore, these substances seemed to increase forced locomotor activity [36]. In the same way, 3-FMC and 4-FMC increased mice locomotor activity, as shown by Marusich et al. [37] and confirmed by Wojcieszak et al., who described effects on locomotor activity due to 3-fluoromethcathinone and methcathinone consumption [38]. In addition, Nadal-Gratacòs et al. confirmed that ring-substituted PVP derivates stimulated locomotor activity in i.p.-injected mice [39]. In the same way, i.p. injections of N-ethyl-pentedrone in mice caused an increase in spontaneous horizontal activity [40]. Ray et al. studied α-pyrrolidinopropiophenone (α-PPP) effects on male mice, by administering 4 unit doses of 80 mg/kg of α-PPP every 2 h, for an 8 h period. After treatment they noted a decrease in mice exploratory activity (through the NOR test); they also found that this cathinone had a locomotor stimulant effect similar to cocaine on mice [41]. In a different study conducted by Centazzo et al., treatment with subcutaneous methylone (6, 12, or 24 mg/kg) significantly altered male mice’s locomotor behavior which was positively correlated with the brain concentration of methylone [42]; also i.p. methylone injections in rats caused an increase of locomotor activity [43].

Several studies have focused on the effects caused by methylenedioxypyrovalerone and these indicated that methylenedioxypyrovalerone enhanced mice locomotor activity, even at low concentrations [37,44,45], acting on horizontal or vertical activity [46]; moreover, methylenedioxypyrovalerone i.p. injections can cause interoceptive dose-time dependent effects [47] and increase mice motility 30 min after the first injection [48]. Methylenedioxypyrovalerone induced self administration in rats [49] and compromised discrimination between familiar and unfamiliar objects [50]; it compromised NOR [50,51] and its ingestion caused social play behavior repression in rats [52]. Treatment with α-PVT can induce self administration in rats [53] and stimulate locomotion in mice [54]. N-ethylpentylone injected in rats caused hyperlocomotion and acted in a similar way to methamphetamine [55,56].

Souders et al. investigated the effects of pyrovalerone in *Danio Rerio* larvae. Pyrovalerone was directly administrated to zebrafish larvae using a special 96-well plate precoated with pyrovalerone. This study reported changes in locomotor activities, showing hyperactivity and higher movement after pyrovalerone administration [57]. This result was observed also in adult zebrafish after α-pyrrolidinopentiophenone exposure [58].

In conclusion, it was confirmed that the effect on locomotory activity is common across different types of synthetic cathinones [23,30,33,36,41,42,57,59,60,61,62].

It is known that SCs can cross the blood-brain barrier. In this regard, Fabegrat-Safont et al. investigated the capacity of 13 different cathinone-derived compounds to cross the BBB. Their paper showed that an increased polarity and the presence of fluorine atoms enhanced cathinones’ ability to cross the BBB, as compared to cathinones with less polar N functionalization, long alkyl chain or non-polar aromatic ring, which crossed the BBB with difficulty—probably because crossing the BBB is a carrier-mediated process [63].

Several studies investigated how SCs interacted with neurotransmitters [41,42,64,65,66,67,68,69,70,71,72] and in particular their effects on levels of dopamine (DA) and serotonin (5-HT) in different regions of rat brain. Martìnez-Clemente et al. reported that mephedrone showed affinity for dopamine transporters and could block dopamine and serotonin uptake in the brain [64]. Other studies confirmed that repeated treatment with mephedrone in adolescent rats caused changes in the basal neurotransmitter levels, especially in striatum, nucleus accumbens and frontal cortex. After i.p. injections of methylenedioxypyrovalerone, mephedrone, and methylone in mice, Allen et al. detected an increase of dopamine levels in the substantia nigra and ventral tegumental areas [73]. Kamińska et al. found an increase of extracellular serotonin levels in nucleus accumbens and frontal cortex and that the ingestion of repeated doses of mephedrone in adolescent mice caused single and double-stranded DNA breaks in the frontal cortex in adulthood [65,74]. Other studies, carried out on mephedrone-treated mice, upheld a loss in the dopamine reuptake in striatum [66,67] and in frontal cortex caused by a decrease in the density of dopamine transporters in these tissues [68] and a decrease in serotonin transporter function in striatal and hippocampal synaptosomes [66,68], amygdala and prefrontal cortex. Studies on mephedrone enantiomers showed that R-mephedrone was more selective for dopamine transporters but was less efficient in serotonin release than S-mephedrone [75]. On the other hand, α-pyrrolidinopentiophenone (α-PVP) increased serotonin levels only in the hypothalamus and increased 3,4-dihydroxyphenylacetic acid (DOPAC, metabolite of dopamine) levels in the amygdala [67]. Other research confirmed that α-PVP was a DAT and NET inhibitor [76,77,78] and caused alteration of dopamine levels in hypothalamus, thalamus and striatum [79]. Ray et al. demonstrated that treatment with α-PPP reduced serotonin levels in striatum in male mice, by examining brain sections of treated mice [41]. Despite data about the effect of mephedrone on serotonin levels, Angoa-Perez et al. demonstrated that it did not cause damage to serotonin nerve endings in female mice and hypothesized that mice were not subject to serotonin nerve ending damage [69].

Comparable studies have been undertaken regarding the effects of methylone, methylenedioxypyrovalerone and α-pyrrolidinopentiophenone.

In particular, it is known that methylone accumulates in the brain of mice in a non-linear manner [42]. Lopez-Arnaw et al. identified a loss of dopamine transporters in the frontal cortex after treatment of mice with methylone, causing a dose dependent neurotoxicity; damage to serotonin and dopamine nerve terminals in the frontal cortex and hippocampus of mice was found after administration of 4 doses of methylone in one day [70]. In addition, methylone or mephedrone swallowed with methamphetamine seemed to induce damage to dopamine (DA) nerve terminals, while MDPV seemed to protect DA nerve endings from combined treatment with amphetamine [71]. Cameron et al. studied methylenedioxypyrovalerone and mephedrone effects on Xenopus oocytes and they found that these substances had different behavior; indeed, methylenedioxypyrovalerone acted as a dopaminergic reuptake inhibitor as opposed to mephedrone, which was a dopamine releasing agent [80]. However, it emerged that methylone or MDPV did not affect levels of 5-HT in the striatum in male mice [72]. Methylenedioxypyrovalerone caused neurodegeneration in different brain areas [51]; in particular, Colon-Perez et al. demonstrated a decrease in brain connectivity in the prelimbic area and nucleus accumbens [81].

Studies on different cell lines investigated the cytotoxic effect of various SCs. Some studies demonstrated that 3-fluoromethcathinone treatment of immortalized mouse hippocampal cell line HT22 caused a dose dependent decrease in the distribution of cells [82]. Rosas-Hernandez et al. identified effects of MDPV on bovine brain microvascular endothelial cells (bBMVECs). In particular, the MDPV treatment caused a decrease of cellular proliferation and an increase in ROS production, resulting in the disruption of the endothelial cell monolayer and loss of the BBB properties of the cells [83].

In conclusion, the reviewed synthetic cathinones can cross the brain-blood barrier and act like psychostimulants in terms of their effects on neurotransmitter levels in different regions of the brain.

**Table 1 ijms-24-06230-t001:** Characteristics of eligible animal studies in vivo and in vitro.

Study Design	Tested/Studied/Reviewed Synthetic Cathinones	Model	Treatment	Results	Study
Research article In vivo study	Mephedrone	Male *Sprague*- *Dawley rats*	Injections (4 × 10 or 25 mg/kg s.c. per injection, 2-h intervals) Self administration: 0.24 mg per 10 µL infusion	Multiple doses of mephedrone caused a decrease in dopamine levels in the striatal and of serotonin levels in the hippocampus synaptosomes and a reduction of serotonin transporters’ functionality.	(2011) [66]
Research article In vitro study	Mephedrone	Isolated synaptosomes or tissue membrane preparations from male *Sprague-* *Dawley rat* cortex or striatum	Incubation in tube with mephedrone (10−8 to 10−4 M)	Mephedrone shown affinity to the DA transporter; only high concentrations of mephedrone inhibited the uptake of dopamine through the serotonin transporter.	(2012) [64]
Research article In vitro study	3-Fluoromethcathinone, 4-fluoromethcathinone, methylenedioxypyrovalerone, mephedrone, methedrone, and methylone	Male *ICR mice*	I.p. injections of cocaine (10–42 mg/kg), methamphetamine (1–10 mg/kg), methylenedioxypyrovalerone (1–30 mg/kg), of, 4-fluoromethcathinone and methedrone (10–56 mg/kg), and 3- Fluoromethcathinone, mephedrone, and methylone (3–56 mg/kg).	All compounds increased locomotor activity; methylenedioxypyrovalerone caused effects at lower concentration.	(2012) [37]
Research article In vivo study	Mephedrone	Young adult male *Listerhooded rats*	i.p. cathinone (1 or 4 mg/kg), mephedrone (1, 4 or 10 mg/kg) or MDMA (10 mg/kg) two days x week (3 weeks) or single acute injection (for neurochemical analysis)	Cathinone, mephedrone and 3,4-methylenedioxy methamphetamine caused hyperactivity in rats. There was a deficit in visual recognition memory and impaired NOD.	(2013) [30]
Research article In vitro study	Mephedrone and 3,4-methylenedioxypyrovalerone	*Xenopus* oocytes	At −60 mV, mephedrone and 3,4-methylenedioxypyrovalerone (0.01–10 µM)	Mephedrone and MDPV had different behavior; while mephedrone acted like a dopamine releasing agent, 3,4-methylenedioxypyrovalerone acted like a dopamine reuptake inhibitor.	(2013) [80]
Research article In vivo study	3,4-methylenedioxypyrovalerone	Male *NIH Swiss mice*	i.p. injections of 3,4-methylenedioxypyrovalerone (0.3, 1.0, 3, 10, 30 mg/kg)	Mice could discriminate methylenedioxypyrovalerone from saline in a pharmacology specific manner. 3,4-methylenedioxypyrovalerone consumption caused interoceptive effects in mice, that were dose and time dependent.	(2013) [47]
Research article In vitro study	3-Fluoromethcathinone	Immortalized mouse hippocampal neuronal HT22 cell line	Incubation with 1-2-4 mM 3-Fuoromethcathinone	Cytotoxic effects on HT22 cell line.	(2014) [82]
Research article In vivo study	Mephedrone	Female *C57BL/6* *mice*	i.p. injections binge-like regimen 4 doses of 20 mg/kg (2 h interval) + methamphetamine (4× 5 mg/kg) or MDMA (4× 20 mg/kg) alone or in combination	Mephedrone did not caused persistent deficit in serotonin nerve endings.	(2014) [69]
Research article In vivo study	Mephedrone	Male Swiss *CD-1 mice*	Subcutaneous injections Schedule 1: 4 doses of saline (5 mL/kg) or mephedrone (50 mg/kg) 2 h interval. Schedule 2: 4 doses of saline or mephedrone (25 mg/kg) 2 h intervals. Schedule 3: 3 doses of saline or mephedrone (25 mg/kg) 2 h interval (×2 consecutive days)	Mephedrone treatment caused aggressive behavior in mice that resulted in self-harm; mephedrone induced decrease in dopamine transporters density in striatum and frontal cortex membrane in mice and a loss of serotonin transporters in the hippocampus.	(2014) [68]
Research article In vivo study	Methylone	Male Swiss *CD-1 mice*	Subcutaneous injections: Treatment A 3 doses (25 mg/kg) 3.5-h intervals ×2 consecutive days. Treatment B 4 doses (25 mg/kg) 3-h intervals in 1 day.	Methylone induced dose-dependent neurotoxicity in mice and impairment of serotonin and dopamine terminals in hippocampus and frontal cortex.	(2014) [70]
Research article In vivo study	Methylenedioxypyrovalerone	*C57BL/6J mice*	i.p. injection of 10 mg/kg b.wt. of methylenedioxypyrovalerone or vehicle	Methylenedioxypyrovalerone treatment caused an elevated mice motility after 30 min, with a major increase after 60 min. Mice rolled without a cause.	(2014) [48]
Research article In vivo study and in vitro study	4-methyl-N-ethylcathinone, 4-methyla-pyrrolidinopropiophenone	Male *Sprague-Dawley rats* (+synaptosomes)/*Xenopus laevis* frogs (oocyte)	In vivo: i.v. injections of 4-methyl-N-ethylcathinone, 40-methyla-pyrrolidinopropiophenone (1, 3 mg/kg) In vitro: 4-methyl-N-ethylcathinone, 4-methyla-pyrrolidinopropiophenone (0.3, 1, 3, 10, 30, 100 uM)	Mephedrone consumption caused an increase in extracellular serotonin, and had little effect on dopamine. 4-methyla-pyrrolidinopropiophenone increased extracellular dopamine selectively and had stimulant effect on locomotor activity. Mephedrone was a non-selective transporter blocker, while 4-methyla-pyrrolidinopropiophenone was selective for hDAT	(2015) [31]
Research article In vivo and in vitro study	R-mephedrone, S-mephedrone	Male *Sprague-Dawley rats* and synaptosomes	In vitro assay: R-MEPH (31.07 nM, 1.47 uM), S-MEPH (74.23 nM, 60.91 nM), racemic (54.31 nM, 83.28 nM) Locomotor exprements: i.p. of saline, R-MEPH, S-MEPH day 1 15 mg/kg, day 2–6 30 mg/kg, day 7 15 mg/kg, after 10 days 15 mg/kg. ICSS: i.p. R-MEPH, S-MEPH (1.0, 3.2, 10 mg/kg). CPP experiments: i.p. R-MEPH, S-MEPH (5, 15, 30 mg/kg).	R-mephedrone appeared more selective for dopamine transporters than its enantiomer, while it was weaker in release of serotonin. Acute exposure to R-mephedrone caused more repetitive movement compared to S-Mephedrone. R-mephedrone induced dose-dependent place preference and higher ICSS facilitation.	(2015) [75]
Research article In vivo study	Methylone, 3,4-methylenedioxypyrovalerone, mephedrone	Female *C57BL/6* *mice*	4 i.p. injections of methylone (30 mg/kg), 3,4-methylenedioxypyrovalerone (30 mg/kg), mephedrone (40 mg/kg) (2 h interval).	Methylone and mephedrone enhanced the damage to dopamine nerve terminals caused by methamphetamine. 3,4-Methylenedioxypyrovalerone protected against neurotoxicity.	(2015) [71]
Research article In vivo study	Methcathinone, pentedrone, pentylone, 3-fluoromethcathinone, and 4-methylethcathinone	Male Swiss *Webster mice*/Male *Sprague-Dawley rats*	Injections of methcathinone, 3-fluoromethcathinone (0.3, 1, 3, 10 or 30 mg/kg); pentylone, 4-methylethcathinone (3, 10, 30 or 100 mg/kg); or pentedrone (1, 2.5, 5, 10 or 25 mg/kg)	All tested drugs caused increase in locomotor activity, at different concentrations, and they showed stimulant effects similar to cocaine, methamphetamine or both.	(2015) [59]
Research article In vitro study	3,4-Methylenedioxypyrovalerone	Bovine Brain Microvascular Endothelial Cells (bBMVECs)	bBMVECs were treated with 0.5 mM to 2.5 mM of methamphetamine, MDMA or 3,4-Methylenedioxypyrovalerone ×24 h.	Methylenedioxypyrovalerone induced cytotoxicity at low concentration, and it inhibited cellular proliferation, so cells lost their blood-brain barrier properties.	(2016) [83]
Research article In vivo study	Mephedrone, flephedrone, clephedrone, mephedrone, brephedrone and methedrone	Adult male *Sprague-Dawley rats*	i.p. injections of saline vehicle, amphetamine (0.1–1.0 mg/kg), fenfluramine (1.0–3.2 mg/kg), mephedrone (0.32–3.2 mg/kg), flephedrone, clephedrone, mephedrone, brephedrone, (1–10 mg/kg), and methedrone (3.2–32 mg/kg)	Mephedrone, flephedrone, and clephedrone increased levels of DA and 5-HT in the nucleus accumbens; the five substituent compounds seemed have more selectivity for DA than 5-HT.	(2016) [74]
Research article In vivo study	4-methylethcathinone	Male *Sprague Dawley rats*	i.p. injections of received 4-methylethcathinone (1, 3, or 10 mg/kg) or methamphetamine (1 mg/kg)	Acute exposure to 4-methylethcathinone enhanced locomotor activity, although chronic exposure caused sensitization, lower than with methamphetamine. 4-methylmethcathinone produced a discriminative stimulus effect, in a similar way to methamphetamine and caused CPP. Chronic consumption of 4-MEC could enhance exploration and decrease anxiety.	(2016) [32]
Research article In vivo study	3,4-Methylenedioxypyrovalerone, methylone	Male *Sprague-Dawley rats*	intravenous (i.v.) self administration 0.03 mg/kg/inj for MDPV, 0.3 or 0.5 mg/kg/inj for methylone, and 0.5 mg/kg/inj for cocaine. Injections of 3,4-Methylenedioxypyrovalerone (0.1, 0.3 mg/kg), methylone (1.0, 3.0 mg/kg)	The acquisition of self administration was faster for MDPV than methylone; methylone is less powerful than MDPV and did not show a dose-dependent effect.	(2016) [49]
Research article In vivo study	4-methylmethcathinone and 3,4-methylenedioxypyrovalerone	Male *Sprague-Dawley rats*	i.p. injections of saline, 3,4-methylenedioxypyrovalerone (0.5 mg/kg), 4-methylmethcathinone or mixtures of 0.5 mg/kg 3,4-methylenedioxypyrovalerone + 4-methylmethcathinone (0.5, 1.0, or 2.0 mg/kg)	3,4-methylenedioxypyrovalerone in combination with 4-methylmethcathinone and 3,4-methylenedioxypyrovalerone caused increase in locomotor activity in mice. 4-methylmethcathinone and 3,4-methylenedioxypyrovalerone interacted with different sites but both conditioned monoaminergic functions.	(2016) [44]
Research article In vivo study	3,4-methylenedioxypyrovalerone	Adult male *Long–Evans rats*	i.p. injections of 3,4-methylenedioxypyrovalerone (0.3, 1.0, or 3.0 mg/kg)	Consumption of a higher dose of 3,4-methylenedioxypyrovalerone caused a decrease in connectivity, in particular in the nucleus accumbens and prelimbic area. Cis-flupenthixol in combination with 3,4-methylenedioxypyrovalerone did not change the reduction of connectivity, but reduced the locomotor stimulant effect of 3,4-methylenedioxypyrovalerone (so it is only partially a dopaminergic stimulation).	(2016) [81]
Research article In vitro study	Methylone, 3,4-methylenedioxypyrovalerone	Female *C57BL/6J mice*	i.p. 4 injection of MDMA 15 or 30 mg/kg, methylone 20 mg/kg, 3,4-methylenedioxypyrovalerone 1 mg/kg; or in combination: methylone/MDMA 20/15 mg/kg, MDPV/MDMA 1/15 mg/kg (2 h interval).	Methylone or methylenedioxypyrovalerone alone did not cause neurotoxic effects in striatum. They mitigated astrogliosis induced by 3,4-methylenedioxymethamphetamine.	(2017) [72]
Research article In vivo study	2-cyclohexyl-2-(methylamino)-1-phenylethanone, 2-(methylamino)-1-phenyloctan-1-one	Male *ICR mice* Male *Sprague-Dawley rats*	i.p. MACHP, MAOP (3 or 30 mg/kg), METH (1 mg/kg) or saline for 7 days; i.p. MACHP, MAOP (3 or 30 mg/kg), METH (1 mg/kg) or saline for 7 days	MACHP and MAOP induced locomotor sensitization effects; in the striatum they could decrease DAT expression.	(2017) [62]
Research article In vivo study	α-pyrrolidinopenthiophenone	*C57BL/6J (B6) mice*/male *Sprague-Dawley rats*	CPP: mice i.p. injections of α-pyrrolidinopenthiophenone (3, 10, 30, 50 mg/kg) Intravenous SA procedure: rats α-pyrrolidinopenthiophenone (0.1, 0.3, 1 mg/kg/inf 2 h session) Discrimination: rats α-pyrrolidinopenthiophenone (1.78, 3.2, 5.6, 10 mg/kg)	α-pyrrolidinopenthiophenone induced place preference in mice and induced self administration in rats; the discriminative capability induced was similar to cocaine and methamphetamine.	(2017) [53]
Research article In vivo study	Pentylone, pentedrone, α-pyrrolidinohexiophenone, methylone	*Wistar* (Charles River, New York) *rats*	Male rats: pentedrone, α-PPP and pentylone (0.0, 0.5, 1.0, 5.0, 10.0 mg/kg, i.p.), Female rats (IVSA): α-PVP (0.05 mg/kg/inf) or pentedrone (0.2 mg/kg/inf); different cathinones (0.0, 0.025, 0.05, 0.1, 0.3 mg/kg/inf). Follow-up study:0.0125, 0.025, 0.1 mg/kg/inf of α-PVP and α-PHP	Increased locomotor activities after the administration of all drugs. Pentedrone and pentylone function as reinforcers in the self-administration model.	(2018) [33]
Research article In vivo study	Mephedrone	Male *Wistar-Han* *rats*	Doses of 5 mg/kg for 8 days	Exposure to mephedrone in adolescent rats produced changes in the basal neurotransmitter levels in striatum, nucleus accumbens and frontal cortex. Mephedrone probably induced neurotoxicity in the cortical brain region.	(2018) [65]
Research article In vitro study	Methylone, 3,4-methylenedioxypyrovalerone, α-pyrrolidinopentiophenone	Cortical cultures from *Wistar rat* pups	Methylone, 3,4-methylenedioxypyrovalerone (1–1000 mM), α-pyrrolidinopentiophenone (1–300 mM)	3,4-Methylenedioxypyrovalerone and α-pyrrolidinopentiophenone inhibited weighted mean burst rate.	(2018) [84]
Research article In vivo and in vivo study	Methylenedioxypyrovalerone, MDPBP, MDPPP, α-pyrrolidinopenthiophenone, and α-pyrrolidinopropiophenone	Male *Sprague-Dawley rats* and synaptosomes	I.v. injection of MDPBP (0.1 mg/kg/inf), MDPPP, α-pyrrolidinopenthiophenone, α-pyrrolidinopropiophenone (0.32 mg/kg/inf) for 10 daily 90-min sessions and Methylenedioxypyrovalerone, MDPBP, MDPPP, α-pyrrolidinopenthiophenone, and α-pyrrolidinopropiophenone, cocaine (0.32, 0.1 mg/kg/inf) in daily 120-min sessions	The compounds were inhibitors of DAT and NET; their reinforcing effect was higher than the cocaine effect. This capability was related to the inhibitor effect on DAT uptake.	(2018) [76]
Research article In vivo study	α-pyrrolidinopentiophenone, α-pyrrolidinoheptanophenone, α-pyrrolidinooctanophenone	*C57BL/6J* *inbred mice*	S.c. injections. Locomotor activity: PVP (1, 3, 10 mg/kg) or PV8 or PV9 (3, 10, 15 mg/kg) or methamphetamine (0.3,1, 3 mg/kg). Spontaneous locomotor activity: pretreatment with SCH23390 (0.06 mg/kg),after PVP or methampheamine (3 mg/kg) or PV8 or PV9 (10 mg/kg). Total effects on dopamine and serotonin levels: PVP or PV8 or PV9 (3, 10 mg/kg)	α-pyrrolidinoheptanophenone and α-pyrrolidinooctanophenone exposure in mice caused stimulation of locomotion and increase of extracellular DA levels in striatum, these effects were higher when α-pyrrolidinopentiophenone was administered. They also increased extracellular 5-HT levels	(2018) [54]
Research article In vivo study	3,4-methylenedioxypyrovalerone	Adult male *Sprague-Dawley rats*	I.v. infusion of 3,4-methylenedioxypyrovalerone, 0.03 mg/kg/inf in 96-hr sessions	Consumption of 3,4-methylenedioxypyrovalerone extended in time caused deficit in NOR memory, causing degeneration of perirhinal and entorhinal cortices, while 3,4-methylenedioxypyrovalerone self administration did not affect SOR memory. Repeated consumption of 3,4-methylenedioxypyrovalerone could compromise cognitive function and induce neurodegeneration in defined brain regions.	(2018) [51]
Research article In vivo study	N-ethylpentylone	Male *Sprague-Dawley rats*	I.p. injections of N-ethylpentylone (5, 20 or 50 mg/kg) or Saline (negative control) or Methamphetamine (5 mg/kg—positive control) ×7 days and after 15 days from day 7	Administration of acute doses of N-ethylpentylone enhanced rats’ locomotor activity. Only after repeated administration of 5 mg/kg doses of N-ethylpentylone was there increased locomotor activity. N-ethylpentylone could cause anxiolytic-like effects.	(2019) [56]
Research article In vivo study	α-Pyrrolidinopentiophenone, 4-methylmethcathinone	Adult female *Sprague-* *Dawley rats*	α-pyrrolidinopentiophenone (0.1 mg/kg/infusion), 4-methylmethcathinone (0.5 mg/kg/infusion), or saline	α-pyrrolidinopentiophenone and 4-methylmethcathinone caused changes in neurotransmitter levels in amygdala, hypothalamus and prefrontal cortex.	(2019) [67]
Research article In vitro study	Methylone, 3,4-methylenedioxypyrovalerone, α-pyrrolidinopentiophenone	Cortical cultures from *Wistar rat* pups	Methylone, 3,4-methylenedioxypyrovalerone (1–1000 mM), α-pyrrolidinopentiophenone (1–300 mM)	Methylone inhibited neuronal activity during acute Exposure; once washed out there is a full recovery.	(2019) [85]
Research article In vivo study	Pyrovalerone	*Danio Rerio*	Doses from 245 ng/mL to 24.5 μg/mL (1–100 μM)	Pyrovalerone treatment induced locomotor changes and anxiolytic behavioral in zebrafish larvae.	(2019) [57]
Research article In vivo study	α-Pyrrolidinopentiophenone	Adult *Danio Rerio*	Acute exposure:1, 5, 25 and 50 mg/L ×20 min Chronic exposure:1, 5 and 10 mg/L ×7 days	Administration of 5, 25, 50 mg/L of α-pyrrolidinopentiophenone in zebrafish caused stimulant-like effects and a stereotypic „side-to-side” swimming. 7 days after chronic exposure or repeated withdrawal, zebrafish showed hypolocomotion.	(2019) [58]
Research article In vivo study	Methcathinone, 3-fluoromethcathinone	Male *C57BL/6J mice*	Locomotor activity: Methcathinone or 3-fluoromethcathinone (1, 3, 10 mg/kg). Studies on D1-dopamine receptor: pretreatment with SCH23390 (0.06 mg/kg) and treatment with methcathinone or 3-fluoromethcathinone (10 mg/kg).	Consumption of methcathinone and 3-fluoromethcathinone caused increase of spontaneous locomotor activity in mice; when D1-dopmine receptors were blocked, this effect was cancelled. The exposure to methcathinone and 3-fluoromethcathinone elicited an important increase of extracellular levels of dopamine and serotonin in the mouse striatum. These effects were like the effects caused by methamphetamine in mice.	(2019) [38]
Research article In vivo study	Methylenedioxypyrovalerone	Male *Sprague-Dawley rats*	I.p. injections of methylenedioxypyrovalerone (3 mg/kg)	Methylenedioxypyrovalerone consumption increased horizontal and vertical activity immediately, affected short-term recognition memory, and caused receptor dopamine-mediated memory impairment. Chronic consumption caused sensitization or tolerance to the locomotor effects depending on the route of administration. Dopamine receptors D1 in nucleus accumbens could mediate methylenedioxypyrovalerone effects.	(2019) [46]
Research article In vivo study	3,4-methylenedioxypyrovalerone, mephedrone, and methylone	Young adult male *Swiss Webster mice*	Monoamine levels: i.p. injections of 3,4-methylenedioxypyrovalerone, mephedrone, and methylone (1.0, 10 mg/kg) and cathinone cocktail (1.0, 3.3, 10 mg/kg) Locomotor activity: i.p. injections of 3,4-methylenedioxypyrovalerone, mephedrone, and methylone separated and together (10 mg/kg)	An increase in dopamine levels in nucleus accumbens, striatum, substantia nigra, and ventral tegmental area, is detected after consumption of high doses of 3,4-methylenedioxypyrovalerone and mephedrone, and methylone, an increase of HVA was also found. A combination of drugs induced higher levels of dopamine. The highest dose of combined drugs caused hypolocomotion and immobility that was not caused by the drug administration alone.	(2019) [73]
Research article In vivo and in vitro study	butylone and pentylone	Male-*Sprague-Dawley rats* and synaptosomes	In vivo: i.v. injections of butylone or pentylone (1 mg/kg, after 60 min 3 mg/kg) In vitro: butylone and pentylone (dose range 0.1 nM–10 µM)	Pentylone and butylone had psychostimulant effects. Their administration increased extracellular dopamine levels, but butylone had more effect on serotonin levels. Pentylone stimulated hyperactivity more effectively than butylone and had better locomotor activity stimulant effects and increased stereotypy (butylone did not have this effect). Both substances were inhibitors of DAT and SERT but pentylone was more selective for DAT.	(2019) [34]
Research article In vivo and in vitro study	3,4-methylenedioxypyrovalerone	Rat pheochromocytoma (PC 12) Male adult *Sprague-Dawley rats* (+synaptosomes)	PC12: 0.1 µM of 3,4-methylenedioxypyrovalerone and cocaine Rats: s.c. injections of 3,4-methylenedioxypyrovalerone (1.5 mg/kg) and i.p. injections of cocaine (30 mg/kg) ×5 days, 10 days of withdrawal and 1 administration	Consumption of 3,4-methylenedioxypyrovalerone caused an increase in locomotor activity, after subsequent exposures there was a decrease in the induced locomotor activity. Acute consumption of 3,4-methylenedioxypyrovalerone caused upregulation of DAT, which was reversible and rapid, but more administrations developed sensitization.	(2019) [45]
Research article In vivo study	N-ethylpentylone, dimethylone, dibutylone, clephedrone (4-Chloromethcathinone), and TH-PVP	Male Swiss *Webster mice* Male *Sprague-Dawley rats*	i.p. in a volume of 1 mL/kg in rats and 10 mL/kg in mice of Methamphetamine (0.25, 0.5, 1, 2, 4 mg/kg) N-ethylpentylone (1, 2.5, 5, 10, 25 mg/kg) Dimethylone (2.5, 5, 10, 25, 50 mg/kg) Dibutylone (5, 10, 25, 50 mg/kg) Cocaine (5, 10, 20, 40 mg/kg) Clephedrone (1, 2.5, 5, 10 mg/kg) TH-PVP (10, 25, 50, 100 mg/kg) MDMA (2.5, 5, 10, 25, 50 mg/kg)	All substances, except TH-PVP, increased locomotor activity while TH-PVP decreased locomotor activity. N-Ethylpentylone action was methamphetamine-like, dimethylone and dibutylone cocaine-like, clephedrone MDMA-like.	(2019) [55]
Research article In vivo study	Methylenedioxypyrovalerone	Male *Wistar rats*	Intraperitoneal injection of methylenedioxypyrovalerone (0.025, 0.05, 0.1, 0.25, or 0.5 mg/kg)	Methylenedioxypyrovalerone consumption in juvenile and young adult male rats repressed social play behavior but did not cause sensitization.	(2020) [52]
Research article In vivo study	Methylenedioxypyrovalerone	Male *Sprague–Dawley rats*	i.p. injections of methylenedioxypyrovalerone (single dose 1 mg/kg or 1 mg/kg 3 times/day × 10 days)	After prolonged methylenedioxypyrovalerone exposure, rats’ ability to discriminate between familiar and unfamiliar objects was compromised. Acute exposure to methylenedioxypyrovalerone had effects on the general excitability of the brain, acting on neurotransmitter receptors. It was found that methylenedioxypyrovalerone exposure compromised NOR.	(2020) [50]
Research article In vivo study	3-Chloromethcathinone, 4-chloromethcathinone, 4-fluoro-α-pyrrolidinopentiophenone, 4-methoxy-α-pyrrolidinopentiophenone	Adult male *C57BL/6J* *inbred mice*	Locomotor activity: s.c. injection 3-Chloromethcathinone, 4-chloromethcathinone, 4-methoxy-α-pyrrolidinopentiophenone or 4-fluoro-α-pyrrolidinopentiophenone (5, 10, 20 mg/kg) Motor performance: s.c. with vehicle (0.9% saline), 3-Chloromethcathinone, 4-chloromethcathinone, 4-methoxy-α-pyrrolidinopentiophenone, or 4-fluoro-α-pyrrolidinopentiophenone (10 and 20 mg/kg)	3-Chloromethcathinone, 4-chloromethcathinone, 4-fluoro-α-pyrrolidinopentiophenone and 4-methoxy-α-pyrrolidinopentiophenone stimulated spontaneous horizontal locomotor activity in mice. Only pyrovalerones caused elevation of vertical locomotor activity. After 4-methoxy-α-pyrrolidinopentiophenone treatment only, there was an improvement of the performance of mice on the accelerating rotarod.	(2020) [36]
Research article In vivo study	Buphedrone, butylone, 4-fluoropentedrone, α-pyrrolidinopentiophenone, pentedrone, pentylone, N,N-dimethylpentylone, N-ethyl-pentedrone, N-ethyl-pentylone, 3,4-methylenedioxypyrovalerone, N-ethyl-hexedrone, N-ethyl-4-methylpentedrone, 3,4-dimethoxy-α-PVP	*Sprague*- *Dawley rats*	i.p. injections of the 13 different cathinones (3 mg/kg)	Synthetic cathinones crossed blood-brain barrier and their permeation was related to their polarity. The process used to penetrate the blood-brain barrier was carrier-mediated.	(2020) [63]
Research article In vivo study	α-Pyrrolidinopropiophenone	Male, Swiss *Webster* *mice*	i.p. injection 4 doses (2 h interval) 80 mg/kg	After exposure to α-pyrrolidinopropiophenone, the levels of monoamine neurotransmitters in striatum and in frontal cortex decreased. Treatment with α-pyrrolidinopropiophenone caused weight loss and had locomotor-stimulants effects.	(2020) [41]
Research article In vivo study	α-pyrrolidinovalerophenone (S and R enantiomers)	Male *Sprague-Dawley rats* and synaptosomes	intravenous (i.v.) injections: racemic, S-α-PVP (0.1, 0.3 mg/kg) and R-α-PVP (1.0 and 3.0 mg/kg) Self administration: R,S-α-PVP, S-α-PVP, R-α-PVP, and R-α-PVP (0.03 mg/kg/inj). Cardiotoxicity: R,S-α-PVP and S-α-PVP (dose range 0.3–3.0 mg/kg). 5 additional rats: R-α-PVP (dose range of 3.0–30 mg/kg)	α-pyrrolidinovalerophenone interacted with DAT and NET blocking their uptake. The activity was minimal at SERT; in particular, S-a-PVP was powerful in blocking uptake at DAT while R-a-PVP was more effective on NET. Both enantiomers caused different psychostimulant effects such as an increase in locomotor activity reflected in increased extracellular dopamine levels in the nucleus accumbens.	(2020) [77]
Research article In vivo study	Dipentylone, N-ethylhexedrone, 4-chloroethcathinone, and 4-methyl-α-pyrrolidinohexiophenone	Male Swiss *Webster mice*	Intraperitoneal injections of methamphetamine (0.25, 0.5, 1, 2, and 4 mg/kg), cocaine (5, 10, 20, and 40 mg/kg), dipentylone, N-ethylhexedrone, 4-chloroethcathinone, or 40-methyl-α-pyrrolidinohexiophenone (2.5, 5, 10, 25, and 50 mg/kg)	All compounds increased locomotor activity in a psychostimulant-like way and produced discriminative stimulant effects.	(2021) [61]
Research article In vivo study	Methylone (and metabolites)	Male *Sprague-* *Dawley rats*	16 s.c. injections of methylone (6, 12, 24 mg/kg) or saline	Methylone and MDC penetrated the central nervous system and crossed the blood-brain barrier. After methylone exposure there was an increase in forward locomotion, rearing and patterned sniffing. Administration of methylone caused an acute depletion of brain serotonin due to the release of monoamine transmitters.	(2021) [42]
Research article In vivo study	α-pyrrolidinopentiophenone and 4-methylmethcathinone	Male and female *Sprague-Dawley rats*	Self administration (jugolar catheter): α-pyrrolidinopentiophenone (0.1 mg/kg/inf) or 4-methylmethcathinone (0.5 mg/kg/inf)	Regarding self administration, there was no sex-linked difference in response between the drugs, LgA and ShA. Rats showed escalation of drug intake over time. The escalation was greater for the LgA group. ShA self-administration escalation was higher for 4MMC. There were some sex-linked differences in the neurochemistry. ShA groups showed differences for NE levels in amygdala, hippocampus, pre frontal cortex, and striatum. LgA groups showed differences in 5-HIAA levels (5-HT metabolite). LgA self administration altered DA levels in hypothalamus, thalamus, and striatum. Consumption of both drugs affected 5-HT and 5-HIAA levels.	(2021) [79]
Research article In vivo and in vitro study	α-Aminovalerophenone derivatives: pentedrone, Nethyl-pentedrone, α-pyrrolidinopenthiophenone, N,N-diethyl-pentedrone and α-piperidinevalerophenone	Male Swiss *CD-1 mice*; Male *Sprague-Dawley rats* synaptosomal preparation	i.p. injections (saline 5 mL/kg, pentedrone or, N-ethyl-pentedrone (3, 10 or 30 mg/kg) or N,N-diethyl-pentedrone (3.5, 12.5 or 35 mg/kg), α-pyrrolidinopenthiophenone (1, 3 or 10 mg/kg) or α-piperidinevalerophenone (7.5, 25 or 75 mg/kg) Tubes with 0.125 mL of pentedrone, N-ethyl-pentedrone, N,N-diethyl-pentedrone, α-pyrrolidinopenthiophenone or α-piperidinevalerophenone at different concentrations in HEPES-buffered solution	Dopamine uptake is blocked by all the studied compounds; thus, all the compounds are DAT inhibitors. N-ethyl-pentedrone, N,N diethylpentedrone and α-piperidinevalerophenone showed psychostimulant and rewarding effects.	(2021) [78]
Research article In vivo study	4-Chloromethcathinone and 4-methoxy-pyrrolidinopentiophenone	*DBA/2J mice*	Subcutaneous injections E1: 4-Chloromethcathinone, 4-methoxy-pyrrolidinopentiophenone (5, 10, 20 mg/kg) E2: 4-Chloromethcathinone, 4-methoxy-pyrrolidinopentiophenone (5, 10 mg/kg) E3: 10, 20 mg/kg E4: intermittent treatment for 14 days (4-Chloromethcathinone, 4-methoxy-pyrrolidinopentiophenone 10 mg/kg) followed by 48 h abstinence	Both substances induced a dose-dependent increase of horizontal spontaneous locomotor activity and behavioral sensitization after treatment, but only 4-methoxy-pyrrolidinopentiophenone increased spontaneous vertical locomotor activity. 4-methoxy-pyrrolidinopentiophenone affected CCP (conditioned place preference)	(2021) [60]
Research article In vivo and in vivo study	Eutylone, dibutylone and pentylone	Rat brain synaptosomes/*C57BL/6J mice*	Subcutaneous injections of eutylone, dibutylone, pentylone (3, 10, or 30 mg/kg)	Eutylone inhibited uptake at DAT and norepinephrine uptake at NET and stimulated partial release of serotonin at SERT. All compounds tested caused a dose-dependent stimulation of hyperlocomotion. Eutylone and pentylone are stronger locomotor stimulants than dibutylone.	(2021) [35]
Research article In vivo study	Methylone	Male and female *Sprague-Dawley rats*	5.6, 10 or 18 mg/kg, i.p.	All three doses of methylone increased activity in rats and they showed significant stereotypies after exposure.	(2021) [43]
Research article In vivo study	Ring-substituted -PVP derivatives. 3-Fluoro- α-Pyrrolidinopentiophenone, 4-Fluoro-α-Pyrrolidinopentiophenone, 3-Chloro- α- Pyrrolidinopentiophenone, 4-Chloro-α-Pyrrolidinopentiophenone, 3,4-Dichloro-α-Pyrrolidinopentiophenone, 3-Bromo-α-Pyrrolidinopentiophenone, and 4-Bromo-α-Pyrrolidinopentiophenone	Male Swiss *CD-1 mice*	i.p. injections: 3-Fluoro- α-Pyrrolidinopentiophenone, 4-Fluoro-α-Pyrrolidinopentiophenone, 3-Chloro- α- Pyrrolidinopentiophenone, 4-Chloro-α-Pyrrolidinopentiophenone, 3,4-Dichloro-α-Pyrrolidinopentiophenone, 3-Bromo-α-Pyrrolidinopentiophenone, and 4-Bromo-α-Pyrrolidinopentiophenone (2.5, 10 or 25 mg/kg)	All the tested substances presented selectivity for DAT; meta-halogen-PVP derivatives increased DA uptake inhibition potency and DAT binding affinity more than their para-analogs. They decreased 5-HT uptake inhibition potency and SERT binding affinity in vitro. α-PVP halogen-ring-derivates caused an acute anxiogenic state. All tested compounds stimulated locomotor activity.	(2022) [39]
Research article In vivo study	N-ethyl-pentedrone	Male *OF1 mice*	i.p. injections of N-ethyl-pentedrone Acute dose (1, 3 or 10 mg/kg, i.p.) or repeated injections of these doses (2 ×day, ×5 days)	N-ethyl-pentedrone consumption in mice caused anxiolytic-like effects or depression and decreased social exploration. The consumption of N-ethyl-pentedrone increased the spontaneous horizontal activity and aggressive behaviors in mice. A decrease of dopamine levels was found in mice after 1 mg/kg exposure.	(2022) [40]

Abbreviations: DA, dopamine; i.p., intraperitoneal; MDMA, 3,4-Methylenedioxymethamphetamine; NOD, novel object discrimination; MDPV, 3,4-methylenedioxypyrovalerone; i.v., intravenous; hDAT, human dopamine active transporter; MEPH, mephedrone; 5-HT, serotonin; CPP, conditioned place preference; 4-MEC, 4-methylethcathinone; MACHP, 2-cyclohexyl-2-(methylamino)-1-phenylethanone; MAOP, 2-(methylamino)-1-phenyloctan-1-one; α-PPP, α-pyrrolidinopropiophenone; α-PVP, α-pyrrolidinopentiophenone; α-PHP, α-pyrrolidinohexiophenone; MDPBP, 3’,4’-Methylenedioxy-alpha-pyrrolidinobutiophenone; MDPPP, 3′,4′-methylenedioxy-α-pyrrolidinopropiophenone; s.c., subcutaneous; PV8, α-pyrrolidinoheptanophenone; PV9, α-pyrrolidinooctanophenone; SOR, spatial object recognition; NOR, novel object recognition; HVA, homovanillic acid; TH-PVP, 3’,4’-tetramethylene-α-Pyrrolidinovalerophenone; MDC, 3,4-Methylenedioxycathinone; LgA, long access; ShA, short access; 5-HIAA, 5-hydroxyindoleacetic acid.

### 3.2. Findings on Humans

SCs’ abuse enhances social and sensory experiences (stimulant, hallucinatory and hedonic) in users, such as increased alertness and awareness, high energy, mood lift and feeling of well-being, euphoria, increased motivation, empathy, work productivity and capacity, self-confidence, sociability, talkativeness, hallucination, insomnia, reduced appetite, motor excitement, increased libido, and sexual arousal. However, there are several unpleasant and adverse effects that occur in acute and chronic intoxication or overdose [13,86,87,88,89,90,91,92,93,94,95,96,97].

Unpleasant or adverse effects are linked to the dopaminergic system stimulation. They include agitation, anxiety, cognitive disorders, delusions, visual and auditory hallucinations, aggressive and erratic behavior, paranoia, psychosis, and seizures, which are the most reported neurological and psychiatric symptoms regarding SC-related intoxications or overdoses. Brain-related adverse effects include stroke, encephalopathy, coma, and convulsions [98,99].

On the other hand, the hallucinogenic toxidrome includes disorientation, hallucinations, memory disruption, psychotic episodes, and tachypnoea, in addition to anxiety, paranoia, hypertension and tachycardia, as observed in the toxidrome described earlier [100], and as reported following “bath salts” abuse [101]. The abusers manifested serotonin syndrome characterized by mental status derangements (e.g., agitation, anxiety and confusion), hyperactivity of the autonomic nervous system (e.g., diaphoresis, hypertension, hyperthermia and tachycardia) and neuromuscular abnormalities (e.g., bilateral Babinski sign, hyperreflexia, muscle rigidity, myoclonus and tremor) [102].

In the case of a synthetic cathinone-related lethal intoxication, the abusers develop excited delirium syndrome [103].

There are several studies regarding SC-induced intoxication but only a few papers dealing with fatalities linked to the related brain neurotoxic effects. Regarding this issue, we collected a series of in vitro and in vivo papers about humans and human cell cultures (Table 2 and Table 3). The substances most involved in intoxication are N-ethylcathinone, α- PVP, MDPHP, N-ethylenorpentylone, 3,4 MDPHP (3,4-Methylenedioxy-α-pyrrolidinohexiophenone), MDPV, Flephedrone, while those related to fatalities are MDPHP and 3,4-MDPHP.

#### 3.2.1. In Vitro Studies of Human Cells

Neurotoxic effects of SCs demonstrated in animal cell lines are confirmed in human in vitro studies. The results obtained are summarized in Table 2.

Leong et al. studied the effects of butylone, pentylone and methylenedioxypyrovalerone treatment on SH-SY5Y cells (human neuroblastoma cell line). They proved that SCs administration caused cell death and an increase in ROS production. This condition is reflected in reduced mitochondrial respiration and in an alteration of Ca^2+^ levels in cells. After a molecular analysis, they showed that these substances also promoted apoptosis because they supported caspase 3 and 7 activation [21]. Several other studies on SH-SY5Y confirmed that SCs could reduce cell viability and increase ROS production [104], also interacting with mitochondria and their functionality [105].

Alteration of the blood-brain barrier is a common effect of SCs consumption reflected in in vitro human models. The experiments conducted by Buzhdygan et al. on the effects of mephedrone on human brain microvascular endothelial cells (hBMVEC) showed that the treatment with mephedrone caused a charge-dependent disruption of the endothelial barrier and an increase of barrier permeability [106], demonstrating the ability of mephedrone to regulate the tight junction complex.

In particular, studies on HEK-293 cells showed that SCs acted like DAT inhibitors with different activity levels depending on the type of SC [31,34].

#### 3.2.2. Lethality—Human Toxicology

The results are summarized in Table 3 and in Figure 4.

Ten papers dealt with SC fatalities associated with previous neurological symptoms such as disorientation, hyperthermia, aggressive behavior, rage or fury, psychomotor agitation (shouting, jumping, running, removing clothes), seizures and convulsions. One case dealt with death in utero in a pregnant woman exposed to MDPHP [107]. Stillbirths secondary to drug exposure are rarely described in the literature; however, some case reports described death in utero after exposure to methadone [108] or to cocaine and methamphetamine [109].

The average age in the present case reports was 31 yrs (n = 9) because fetal death was not considered among these. Eight out of ten were male and two were female.

In the cases reported here, SCs were detected and quantified in post-mortem biological samples (n = 8), except for two cases. In the first of these, tests for SC metabolites and for 25INBOMe (5-HTA2 receptor agonist) were negative at hospital admission, while post-mortem GC and MS analyses revealed N-ethylpentylone in the urine [110]; in the second case, routine drug screening was performed post-mortem showing N-Ethylpentylone [111].

The highest SC blood concentration was 4200 ng/mL detected in a fatal Eutylone intoxication from a single dose [112], while the lowest serum concentration was 82 ng/mL [113]. It is noteworthy that a fetal case was associated with an even lower concentration equal to 76 ng/mL [107].

In 8 out of 10 SC fatalities a single cathinone was detected in the biological samples, while in the remaining two fatalities two SCs were detected as main drugs of abuse such as MDPV and methylone [114] and 3,4-MDPHP and α-PHP [107].

In 2 out of the 10 fatalities, no other drugs were detected except for SCs [107,115]; in the remaining fatalities, biological samples revealed more drugs or psychotropic substances (poly drug fatalities).

The prevalence of other drugs associated with the main SCs is summarized in Figure 1. The most represented “accompanying drugs” were EtOH and psychotropic drugs (such as Aripiprazole, Tricyclic antidepressants, Lamotrigine). The evidence of psychotropic drug use in the case of multidrug fatalities underlines the great diffusion of SCs among patients already affected by psychiatric disorders and in therapy.

#### 3.2.3. Toxicology in Human SCs Intoxication

Twenty-seven papers dealt with case reports. The results are reported in Figure 5 and Figure 6.

In 10 out of the 27 cases, the mean age reported at the moment of hospital admission was 28.8 years.

In 6 out of the 27 cases, gender was not reported; the remaining 21 comprised 19 males and 2 females. In particular, one woman was pregnant during the intoxication event.

In 19 out of the 27 cases, the principal SC was identified and detected. In the remaining cases (8) the main SCs responsible for the intoxication were suspected because of the circumstances (participation in a rave party, or personal admission of the patient or of his friend or parent) without being detected in biological samples.

The cathinone found at the highest blood concentration was α-PVP with 890 ng/mL, while the cathinone found at the lowest blood concentration was MDPHP (mean concentration = 30.3).

In 6 out of 24 cases, SCs were the only drug detected and were considered the main agent responsible for the exogenous intoxication. In fact, 18 out of the 27 patients were poly-drug abusers; among them, intoxication with benzodiazepines tied with opioids (+metabolites) as the substances most represented in addition to CSs.

**Table 2 ijms-24-06230-t002:** Characteristics of eligible studies (in vitro studies of human cells).

Study Design	Synthetic Cathinone	Model	Treatment	Results	Study
Research article In vitro study	α-Aminovalerophenone and derivates: pentedrone, N-ethyl-pentedrone, α-PVP, N,N-diethyl-pentedrone and α-PpVP	Transfected (hDAT, hSERT, hOCT) human embryonic kidney (HEK293)	Drugs were tested in increasing concentrations (0.1 M–300 μM); incubation for 1 h at 20 °C	Tested substances were DAT inhibitors with very low affinity for SERT, hOCT-2 and -3.	(2021) [78]
Research article In vitro study	4’-methyl-alphapyrrolidinoexanophenone	SH-SY5Y cells (neuroblastoma cells line)	Incubation with different drug concentrations (15–1000 µM) for 24 h; incubation with 125 and 250 µM of different drugs for 24 h	24 h after the exposure, all compounds induced a loss of viability and oxidative stress in a concentration-dependent manner. 2-Cl-4,5-MDMA activated apoptotic processes, while 3,4-MDPHP induced necrosis. 3,4-MDPHP exposure did not increase expression levels of pro-apoptotic Bax and caspase 3 activity.	(2021) [104]
Research article In vitro study	4-methyl methcathinone (mephedrone)	hBMVEC	Cells were treated with mephedrone at different concentrations (1, 5, 10, 100 µM) for different times	Mephedrone did not induce cytoxicity but caused reduction of barrier properties and induced proinflammatory response.	(2021) [106]
Research article In vitro study	butylone, pentylone, and 3,4-Methylenedioxypyrovalerone	SH-SY5Y	Incubation with all the compounds (1–10 mM) for 24 h	Butylone, pentylone and 3,4-Methylenedioxypyrovalerone caused neurotoxicity; they enhanced ROS production which caused mitochondrial respiratory chain dysfunction.	(2020) [21]
Research article In vitro study	butylone and pentylone	HEK-293 expressing DAT or SERT	Superfusion with butylone or pentylone (300 nM, 1, 10, 30, 100 and 300 μM)	Butylone and pentylone were DAT and SERT inhibitors; pentylone was more efficacious for DAT.	(2019) [34]
Research article In vivo study	3,4-Methylenedioxypyrovalerone, PV9, 2,3-methylenedioxypyrovalerone, pyrovalerone, demethylenylmethyl- MDPV, 3,4-dihydroxypyrovalerone, a-pyrrolidinopentiothiophenone,	SH-SY5Y, Hep G2, and RPMI 2650 cell lines	Incubation with 100, 200 and 300 µM of drugs	Pyrovalerone, 3,4- methylenedioxypyrovalerone and 2,3-methylenedioxypyrovalerone decreased mitochondrial activity. PV9 and α-PVT had powerful cytotoxic effects.	(2016) [105]
Research article In vitro study	4-methyl-N-ethylcathinone (4-MEC) and 4-methyla- pyrrolidinopropiophenone	HEK-293 expressing DAT or SERT	Incubation with 10 µM 4-methyl-N-ethylcathinone and 10 µM and 4-methyla- pyrrolidinopropiophenone ePPP	4-methyl-N-ethylcathinone acted as SERT substrate and DAT blocker. 4-methyla-pyrrolidinopropiophenone acted as a DAT and SERT inhibitor, and was more powerful for DAT.	(2015) [31]

Abbreviations: α-PVP, α-pyrrolidinopentiophenone; α-PpVP, α-piperidinevalerophenone; hDAT, human dopamine active transporter; hSERT, human Serotonin Transporter; 2-Cl-4,5-MDMA; 6-chloro-methamphetamine; 3,4-MDPHP, 3’,4’-Methylenedioxy-α-pyrrolidinohexiophenone; hBMVEC, human brain microvascular endothelial cells; PV9, α-pyrrolidinooctanophenone; α-PVT, α-Pyrrolidinopentiothiophenone.

**Table 3 ijms-24-06230-t003:** Characteristics of eligible studies—neurological intoxication by SCs and fatalities in human cases.

Study Design	ScS and Matrices and Concentration	Other Drugs Detected	Age, Sex, Clinical Description	Methods of Analyses	Neurological Symptoms	Study
			Intoxication Cases			
Case Series	N-ethylcathinone: Blood = 519 ng/mL	Amphetamine: Blood = 90 ng/mL	Unknown age, M, took ‘crystal’ showing depressed mood, elevated heart rate but after 5 h was drawn	HPLC and LC-MS	acute neurological toxicity	(2022) [116]
Case Series	α-PVP: Blood = 890 ng/mL	Midazolam: Blood = 10 ng/mL	33 yrs, M, i.v. occurred: psychomotor agitation, elevated heart rate, incapacity to walk, involuntary movements	HPLC and LC-MS	acute neurological toxicity	
Case Series	MDPHP: SERUM = 32 ng/mL	EtOH: SERUM (−) THC: SERUM < 0.6 ng/mL DIAZEPAM: SERUM = 54 ng/mL NORDAZEPAM: SERUM = 110 ng/mL	M, aggressive behavior against authorities; impaired balance, delayed physical response	GC–MS (urine); LC–MS-MS (quantitative analysis)	acute neurological toxicity	(2020) [117]
Case Series	MDPHP: SERUM = Approx. 140 ng/mL	EtOH: SERUM = 0.44 mg/mL; THC: SERUM = 0.8 ng/mL; Buprenorphine: SERUM = 3.2 NG/ML	M, riding bicycle under the influence of drugs and alcohol, showed impaired balance, delayed physical response, behavior inconspicuous	GC–MS (urine); LC–MS-MS (quantitative analysis)	acute neurological toxicity	
Case Series	MDPHP: SERUM = 3.3 ng/mL	Amphetamine: SERUM = 21 ng/mL; Cocaine: SERUM< 5 ng/mL; Benzoylecgonine: SERUM = 130 ng/mL; Diazepam: SERUM < 53 ng/mL; Nordazepam: SERUM < 43 ng/mL; Clonazepam: SERUM = 20 ng/mL; 7-Aminoclonazepam: SERUM = 7.0 ng/mL; Buprenorphine: SERUM = 1.0 ng/mL	M, traffic accident while driving under the influence of drugs; delayed physical response, behavior and mood inconspicuous	GC–MS (urine); LC–MS-MS (quantitative analysis)	acute neurological toxicity	
Case Series	MDPHP: SERUM = 3.7 ng/mL	EtOH: SERUM (−); α-Pyrrolidinohexanophenone: SERUM = 32 ng/mL; Dextromethorphan: SERUM = 10 ng/mL	Burglary, stated to be under the influence of heroin; delayed physical response, erratic behavior	GC–MS (urine); LC–MS-MS (quantitative analysis)	acute neurological toxicity	
Case Series	MDPHP: SERUM = 16 ng/mL	EtOH: SERUM (−); Levomethadone: SERUM = 140 ng/mL; EDDP: SERUM = 20 ng/mL; Diazepam: SERUM = 72 ng/mL; Nordazepam: SERUM = 45 ng/mL; Clonazepam: SERUM = 41 ng/mL; 7-Aminoclonazepam: SERUM = 24 ng/mL	Patient known as opiates/opioids and BS user; coma after acute poly-intoxication; admitted at ED with loss of consciousness, respiratory insufficiency; recovery of consciousness after naloxone	GC–MS (urine); LC–MS-MS (quantitative analysis)	acute neurological toxicity	
Case Series	MDPHP: SERUM = 21 ng/mL	EtOH: SERUM (−); Morphine: SERUM = 34 ng/mL; Diazepam: SERUM = 53 ng/mL;	Poly-intoxication; consumed ‘FLEX’; impairment of consciousness and behavior, suicidal thoughts. After admission to ED, spontaneously mind-conscious and oriented	GC–MS (urine); LC–MS-MS (quantitative analysis)	acute neurological toxicity	
Case Series	MDPHP: SERUM = 8.7 ng/mL	EtOH: SERUM (−); Cocaine (benzoylecgonine): SERUM NEGATIVE (23 ng/mL) Morphine: SERUM = 5.0 ng/mL Levomethadone: SERUM < 50 bg/mL; Cannabinoids: URINE= (+)	Previous history of heroin addiction, was found unconscious; vigilance reduction, asystole; cardiopulmonary resuscitation with return of spontaneous circulation; multiple organ failure	GC–MS (urine); LC–MS-MS (quantitative analysis)	acute neurological toxicity	
Case Series	MDPHP: SERUM = 11 ng/mL	Diazepam: SERUM = 10 ng/mL; EtOH: SERUM = 0.10 mg/mL Nordazepam: SERUM = 56 ng/mL; Clonazepam: SERUM = 56 ng/mL; 7-Aminoclonazepam: SERUM = 37 ng/mL	Aggressive behavior and mental confusion; suspicion of drug-induced acute psychosis; taken to the emergency department by the police	GC–MS (urine); LC–MS-MS (quantitative analysis)	acute neurological toxicity; psychosis	
Case Series	MDPHP: SERUM = 37 ng/mL	EtOH: SERUM (−) Levomethadone: SERUM 250 bg/mL; α-Pyrrolidinohexanophenone: SERUM = 1.0 ng/mL	Found in the street with agitation and mental confusion, aggressive behavior	GC–MS (urine); LC–MS-MS (quantitative analysis)	acute neurological toxicity	
Case Series	N-ethylnorpentylone: SERUM = 7 ng/mL; URINE (+)	initially negative for drug	18 yrs, M, brought to ED from a rave party; he displayed signs of several injuries, tachycardia, mydriatic pupils, psychomotor agitation, neurological depression	GC-MS (screening); LC-MS/MS (quantification)	acute neurological toxicity	(2019) [115]
Case Series	N-ethylnorpentylone.	MDMA (+); No cannabinoids, benzodiazepines or GHB were detected.	26 yrs, F, brought by ambulance to ED after having taken part in a drug-party; found unconscious in her apartment with sphincter release; at the ED she was confused, sleepy, with tongue injuries suggestive of intentional bite, disconnected speech, visual hallucinations. TX: hydrated and stayed under observation for 24 h	GC-MS (screening); LC-MS/MS (quantification)	acute neurological toxicity	
Case Series	N-ethylnorpentylone: serum = 19 ng/mL	EtOH: Blood = 0.8 g/L; MDMA: serum = 54 ng/mL MDMA, caffeine, cotinine (nicotine metabolite); NO LSD	19 yrs, M, stayed at rave party ingesting for 14 h (drugs and alcohol). At ED: conscious, oriented, agitated, palpitations, tachycardia, BP = 123/89 mmHg; 37.3 °C; Cr higher values	GC-MS (screening); LC-MS/MS (quantification)	acute neurological toxicity	
Case Series	N-Ethylnorpentylone: SERUM = 149 ng/mL; urine (+)		35 yrs, M, consumed alcohol and other drugs for 2 consecutive days; was found unconscious, neurological depression, anisocoria, and intubation; a CT brain was negative; after 6 h evolution in neurogenic shock, decerebration. CT neck detected vertebral artery dissection affecting the brainstem: diagnosis of brainstem hemorrhage	GC-MS (screening); LC-MS/MS (quantification)	Hypertensive encephalopathy	
Case Series	N-ethylnorpentylone: SERUM = 61 ng/mL		26 yrs, M, previous history of mental disorders, presented symptoms of psychosis, paranoia, sleeplessness, inconsistent speech (talking about wars, guns, Nazism). TX: sedated, hydrated, in observation	GC-MS (screening); LC-MS/MS (quantification)	acute neurological toxicity	
Case Report	3,4-MDPHP: MOTHER Blood = 16 ng/mL; MOTHER URINE = 697 mg/mL; α-PHP: MOTHER Blood = traces; MOTHER URINE = 136 ng/mL	no other drugs at screening	21 yrs, F, pregnant in the 36th week of pregnancy, with a history of smoking tobacco and consuming alcohol, was found unconscious; at the ED she showed psychomotor agitation, anxiety, mumbled speech; admitted in Intensive Care Unit; TX: ceftriaxone and metronidazole; 2nd day: fetal demise with delivery of a 2380 g female infant	liquid chromatography with mass spectrometry (LC–MS/ MS)	acute neurological toxicity	(2019) [107]
Case Series: one case fully described	3-MMC: SERUME = 2 ng/mL; URINE = 85 ng/mL;	buprenorphine (+); ethanol metabolites (ethyl glucuronide and ethyl sulfate) (+); diazepam (+)	25 yrs, M, with a history of chronic hepatitis C and alcohol and amphetamine abuse, found unresponsive; previous evening, he had been acting oddly, slightly in confusion. In ED: unconscious, 39 °C; hypotensive, tachycardic, RR = 36/min, pO2 = 96% while receiving oxygen, metabolic acidosis, lactate = 7.4 mmol/L; TX: intubated; CT/MNR brain: cerebral edema and anoxic brain injury		odd behavior, mental confusion	(2015) [118]
Case Series	___	neg (−) other drug research	39 yrs, M, with a history of bipolar disorder and drug abuse; in ED with altered mental status, violent behavior, oriented but soon after he became confused, agitated, requiring restraints; CPK > 47,000 U/L; Cr = 1.83 mg/dL; after he was found unresponsive, respiratory arrest; TX: resuscitated, intubated, intensive care unit; CT brain: anoxic damage	urine drug screen	sympathetic stimulation	(2013) [119]
Case Series	___	amphetamines (+)	42 yrs, M, with a history of depression, anxiety, alcohol dependence, was brought to ED in agitation and aggressive behavior. In ED: somnolent and hallucinating; tachycardic, RR = 25 b/m, 36.6 °C, PO2 = 94%; TX: intubated; ECG: tachycardia; brain CT: no lesions; CPK = 21,155 U/L; Cr = 1.70 mg/dL; TX: hydration, intensive care; CPK = 4706 U/L	urine drug screen	sympathetic stimulation	
Case Series	___	neg (−) other drug research	33 yrs, M, had few hours’ history of chest discomfort and muscle aches after BS; in ED hallucination, CPK = 6600 U/L; Cr = 1.32 mg/dL; ECG = sinus tachycardia; TX: hydration; CPK = 4277 U/L	urine drug screen	sympathetic stimulation	
Case Series	___	cannabinoids (+)	28 yrs, M, with a history bipolar disorder and attention-deficit/hyperactivity disorder, admitted to ED with altered mental status due to BS; in ED: agitation, making inappropriate comments, dry oral mucosa and warm skin; CPK = 13,500 U/L; Cr = 1.51 mg/dL; TX: intravenous fluid hydration; improvement of renal indices; CPK = 3597 U/L	___	sympathetic stimulation	
Case Series	___	cocaine (+)	35 yrs, M, with a history bipolar disorder, admitted to ED because of acute delirious state associated to BS; he was agitated, violent behavior, confused and disoriented; CPK = 2200 U/L; Creatinine = 1.41 mg/dL; TX: hydration;	___	sympathetic stimulation	
Case Report	MDPV: SERUM = 186 ng/mL; URINE = 136 ng/mL Flephedrone: SERUM = 346 μg/mg; URINE = 257 ng/mL; Catechol pyrovalerone and Methylcatechol pyrovalerone [MDPV metabolites]: URINE (+)	tetrahydrocannabinoids: URINE (+); Caffeine: SERUM = 387 ng/mL; URINE = 367 ng/mL;	23 yrs, M, prior psychiatric history, admitted in ED for bizarre behavior, suicidality, hallucinations after reportedly insufflating BS; agitated, visual, tactile, and auditory hallucinations; BP = 133/68 mmHg; HR = 109 bpm; 36.9 °C; RR = 21 bpm; diaphoretic, tachycardic, mydriasis; Cr = 1.10 mg/dL (v.n. 0.67–1.17).	immunoassay; LC-TOF/MS	psychosis and mild sympathomimetic syndrome	(2012) [120]
Case Report	___		31 yrs, M, with history of paranoid delirium after BS exposure, admitted to ED with fearfulness, confusion, hallucination, hyperthermia; in ED was combative, requiring restraints; diagnosis: AKI, hyperkaliemia, rhabdomyolysis	___	excited delirium	(2012) [121]
Case Report		neg (−) other drug research	30 yrs, M, BS used 2 days before, admitted to ED with paranoia, agitation, violent behavior, change in mental status; in ED unresponsive, 150 hr, CPK = 6599 U/L with a peak of 32,880; Cr = 2.1 mg/dL. Diagnosis: AKI, rhabdomyolysis, multiple organ failure, acute distress syndrome; TX: intubation, hydration twice	urine screening	excited delirium	
Case Report	___	benzodiazepine (+)	26 yrs, M, BS used in the previous 3 days, found fearful, diaphoretic, confused, lying face down; at ED combative attitude, diaphoretic, 170 HR; 41 °C, hemoglobinuria (+3), CPK = 6599 U/L; CT thoracic: negative	urine screening	excited delirium	
	**Concentration**	**Other drugs**	**Age, Sex, Clinical description/Neurological symptoms**	**Methods of analyses**	**Autopsy findings and Cause of Death**	**Reference**
			**Fatality Cases**			
Case Report	Eutylone: ED Blood (+); PM cardiac blood (+); pm gastric content (+); pm urine (+); pm fat tissue (+); Eutylone: PM Blood = 4290 (±167) ng/g ± S.D; URINE = 192,000 ng/g ± S.D; PM GASTRIC CONTENT = 2120 (±63.4) ng/g ± S.D; FAT TISSUE = 1310 ng/g ± S.D; ED PERIPHERICAL Blood = 2500 ng/g ± S.D	EtOH: ED Blood (−); PM Blood (−); PM URINE (−); Aripiprazole: PM CARDIAC Blood (+); PM CARDIAC Blood = 49.1 (±8.82) ng/g ± S.D.; URINE = 34.5 ng/g ± S.D.; FAT TISSUE = 358 ng/g ± S.D; ED PERIPHERICAL Blood = 26.7 ng/g ± S.D.	32 yrs M, with a 4-yrs history of schizophrenia treated with 1 dose of aripiprazole 400 mg/month, came out of an apartment in his underwear, shouting, running around, rolling and jumping; 20 min after the first observation of abnormal behavior, the man was found unconscious. In ambulance he was in PEA and in hospital he was pronounced dead.	gas chromatography (EtOH); screening assay;	autopsy: congestion of internal organs; edematous and congested lungs (right = 877.6 gr; left = 743.4 gr). Microscopic: fatty liver; convoluted tubular necrosis in the kidneys without inflammatory cell infiltrates. COD: accidental eutylone intoxication with delirium and agitation	(2022) [112]
Case Report	N-ethylhexedrone: PM Blood = 145 ng/mL;	ADMISSION: benzodiazepines (+); Amphetamine: PM Blood (+); Amphetamine: PM Blood = 12 ng/mL; CANNADINOIDS: PM Blood (+); 11-nor-9-carboxy-Δ9-tetrahydrocannabinol (THC-COOH): PM Blood < LOQ < 5 ng/mL;	21 yrs M, with a history of drug and alcohol abuse, was admitted to ED because of disorientation, aggression, loss of consciousness; clinically: wide pupils; >41 °C, HR > 160 bpm, RR = 20 bpm, BP = 110/60 mmHg, anuria with AKI; TX: benzodiazepine, ICU; after a few hours of hospitalization, the patient died (sudden cardiac arrest).	Admission [screening]; PM [LC–MS-MS]	Autopsy: lung congestion; focal pulmonary edema; swelling and congestion in brain; left ventricular hypertrophy; liver steatosis. COD: not specified	(2021) [122]
One case from case series	N-ethylnorpentylone: Blood = 170 ng/mL	no other drugs	A 32-year-old man attending a rave party displayed psychomotor agitation and aggressiveness, then eventually fainted; the man died in the ambulance on route to the hospital. Sympathomimetic syndrome due to SCs.	GC-MS and gas chromatography with flame ionization detection (GC-FID).	Autopsy: generalized hemorrhage of the pulmonary alveoli, abnormal increase in liver size and absence of urine in the bladder	(2019) [115]
Case Report	N-ethylnorpentylone: URINE (+)	EtOH = 12 mg/Dl; CANNABINOIDS (+); SCs metabolites and 25INBOMe (−);	21 yrs, M, admitted to ED with odd behavior; soon after, he became unresponsive with cardiac arrest; PA = 95/55 mmHg; HR = 126 bpm; RR = 25 bpm; PO2 = 99%; ECG: sinus rhythm with premature atrial complexes, ST depression in inferior leads, QT = 403 ms; K = 6.8 mmol/L, glucose = 28 mg/dL; CPK = 116,550 IU/L, lactic acid 28 mg/dL; AST = 12,374 IU/L; ALT = 7649 IU/L; Cr = 1.70; elevated troponins and acute kidney injury; MR brain = profound cerebral hypoxia. II day: disseminated intravascular coagulation; III day: severe hypotension requiring vasopressors for hemodynamic support; second cardiac arrest 72 h after admission; death. Cerebral hypoxia	gas chromatography and mass spectrometry	Autopsy: COD = drug intoxication with N-ethylpentylone and encephalic hypoxia	(2018) [110]
Case Report	α-propylaminopentiophenone: Blood = 3.2 ± 0.68 μg/g; LIVER = 5.9–6.0 μg/g; BRAIN = 2.1–2.3 μg/g; HV = 4.4 ± 0.52; KIDNEY = 5.4 ± 0.92 μg/g;	Blood: amphetamine, tricyclic antidepressants, Δ9-tetrahydrocannabinol; benzodiazepines;	29 yrs, F, with previous history of depression, alcoholism, and suicide attempt, after having swallowed lots of white powder she claimed not to feel her legs, being unable to walk; convulsions; lost consciousness; fell to the floor; medical emergence resuscitated her; she died in ED. Convulsion and numbness in both legs	HPLC high-performance liquid chromatography coupled with mass spectrometry	Autopsy: cerebral and pulmonary edema; passive congestion of internal organs; COD = acute respiratory distress following α-propylaminopentiophenone intoxication	(2018) [123]
Case Report	MDPV: FB = 0.44 μg/g; HB= 0,50 μg/g; URINE > 5.0 μg/g; GASTRIC > 2.0 μg/g; BILE = 0.88 μg/g; CSF = 0.41 μg/g; LUNG = 0.98 μg/g; KIDNEY = 0.84 μg/g; LIVER = 0.98 μg/g; MUSCLE = 0.58 μg/g; SPLEEN = 0.64 μg/g; BRAIN (average) = 0.37 μg/g; HEART= 0.12 μg/g; HAIR = 11,660 pg/mg METHILONE: URINE (+); HAIR: 11,660 PG/MG	Lamotrigine: Blood (tl); Fluoxetine: Blood (tl); Risperidone: Blood (tl); Ibuprofen: Blood (tl) Benztropine: Blood (tl); Pseudoephedrine: Blood (tl)	39 yrs, M, with a previous history of schizophrenia, depression, drug abuse, was found face up in bed in unresponsive state. Anoxic brain injury	GC-MS after solid-phase extraction and HPLC–tandem mass spectrometry after solid-phase extraction	Autopsy: enlarged heart (430 g) and edematous lungs (right = 950 g, left = 710 g); COD: MDPV intoxication resulting in anoxic brain injury,	(2013) [114]
Case Report	3,4-MDPHP: FETAL Blood = 76 ng/mL; α-PHP: FETAL Blood = 12 ng/mL;	no other drugs at screening	21 yrs, F, pregnant in the 36th week of pregnancy, with a history smoking tobacco and consuming alcohol, was found unconscious in the apartment by her partner; at the ED she showed psychomotor agitation, anxiety, mumbled speech; because of vaginal spotting signs of intrauterine fetal death, she was admitted in Intensive Care Unit; BP = 160/90 mmHg; HR = 130/min; PO2 = 98%; 36.8 °C, blood glucose = 97 mg/dL; TX: benzodiazepines, propofol, intubation; RX chest: lungs-inflammation; TX: ceftriaxone and metronidazole; 2nd day: fetal demise with delivery of a 2380 g female infant (Apgar = 0), no fluid in the peritoneal cavity.	liquid chromatography with mass spectrometry (LC–MS/ MS)	Autopsy: no significant pathological findings; COD: intrauterine fetal death as a result of asphyxia with drug accumulation in brain	(2019) [107]
Case Report	___	admission blood: free morphine (−), 6-acetylmorphine (−), total morphine = 0.04 mg/L; benzodiazepine (−), amphetamine (−), metamphetamine (−), lidocaine (+)	29 yrs, M, with a previous history of bipolar disorder, intravenous drug abuse and abuse of MDMA/cathinones, was seen wandering on the road in agitation, completely nude; he was belligerent, struggled to get up multiple times and urinated on himself. During the transport, he went in PEA; intubated and Advanced Cardiac Life Support protocol was initiated which restored a pulse; in ED: hypotensive and tachycardic; second cardiac arrest with ACLS return of sinus rhythm; in intensive care unit. The diagnosis was drug overdose, cardiac arrest and hyperthermia; then acidemia, profuse bleeding from his nose due to DIC, treated with multiple blood products; 36 h after admission, he went into cardiac arrest for a third time; was pronounced dead after 45 min of unsuccessful resuscitation.	headspace gas chromatography; immunoassay; LC-MS/MS confirmation; gas chromatography mass spectrometry (GCMS)	Autopsy: pleural and peritoneal effusions, enlarged heart, left ventricular hypertrophy; cerebral edema; edematous lungs (right = 850 g, left = 930 g); COD: accidental intoxication by N-ethylpentylone	(2017) [111]
Case Report	Mephedrone: FB = 5.1 mg/L; URINE = 186 mg/L; GASTRIC CONTENT = 1.04 g/L	FB: cocaine = 0.0071 mg/L; Benzoylecgonine = 0.17 mg/L; Methylecgonine= 0.042 mg/L; MDMA: 0.011 mg/L; Oxazepam < 0.01 mg/L; Midazolam = 0.0064 mg/L; HB: Metanephrine (traces); Atropine (traces);	36 yrs, M, was arrested by police after having injured himself severely by smashing windows, in a rage of fury; medical personnel administrated midazolam and naloxone i.v. but the man lost consciousness; after unsuccessful resuscitation, the man died	General unknown screening after solid phase extraction (SPE) (biological fluid); HPLC–UV and GC–MS (for screening); headspace GC–FID mEtOHd (for EtOH);	Autopsy: superficial veins and tendons of the hand injuries (smashing of the window); bruise, hemorrhage, scratches along the body; brain swelling and lung oedema; COD: fatal oral intake of mephedrone and excited delirium	(2011) [124]
Case Report	MDPV: URINE = 670 ng/mL; SERUM = 82 ng/mL	URINE/SERUM: positivity for acetaminophen, caffeine, cotinine, lidocaine, trimethoprim, and MDPV; quetiapine (not detected); TRIMETHOPRIM: URINE = 12 mcg/mL; SERUM= 2.2 mcg/mL	40 yrs, M, with a history of bipolar disorder, after injecting and snorting BS, became aggressive, uncontrollable, delusional, removed all of his clothing and ran outside; police restrained him; in ambulance: HR = 164 bpm; BP = 131/72 mmHg; RR = 24 breaths/min, PO2 = 100% on non-rebreather mask; ECG: sinus tachycardia with widened QTc interval and peaked T-waves; within 5 min of his arrival to ED: bradycardia and cardiac arrest with PEA; post-arrest period included: K = 7.4 mmol/L, Cr = 3.0 mg/dL; salicylate level = 4.1 mg/dL; resuscitation; afterwards hyperthermia, rhabdomyolysis, coagulopathy, acidosis, anoxic brain injury; death	gas chromatography/mass spectrometry (screening) LC/MSMS for confirmatory test; performance thin layer chromatography (trimethoprim)	Autopsy: Exicited delirium	(2012) [113]

Abbreviations: M, male; F, female; yrs, years; α-PVP, α-pyrrolidinopentiophenone; MDPHP, 3′,4′-Methylenedioxy-α-pyrrolidinohexiophenone; FB, femoral blood; EtOH, Ethanol; PEA, Pulseless electrical activity; 25INBOMe, 2-(4-iodo-2,5-dimethoxyphenyl)-N-[(2-methoxyphenyl)methyl]ethanamine; THC, Tetrahydrocannabinol; 3,4-MDPHP, 3’,4’-Methylenedioxy-α-pyrrolidinohexiophenone; MDMA, 3,4-Methylenedioxymethamphetamine; LSD, lysergic acid diethylamide; ED, emergency department; BP, blood pressure; Cr, Creatinemia; α -PHP, α-Pyrrolidinohexanophenone; α-pyrrolidinohexiophenone; TX, treatment; RX, X-rays; 3-MMC, 3-methylmethcathinone; RR, respiratory rate; CT, computed tomography; NMR, nuclear magnetic resonance; CPK, creatine phosphokinase; PO2, partial pressure of O2; ACLS, Advanced Cardiac Life Support; BP, blood pressure; MDPV, 3,4-methylenedioxypyrovalerone; BS, bath salts; AKI, acute kidney injury; PM, post mortem; COD, cause of death; HV, vitreous humor.

All these reported cases demonstrated that, in humans, synthetic cathinones can cross the blood-brain barrier, be identified in the brain and cause damage to tissue.

## 4. Conclusions

The studies demonstrated that the SCs cause severe intoxication and fatalities.

Synthetic cathinones, more commonly known as “bath salts”, are human-made stimulants, chemically related to cathinone, a substance found in the khat plant. Khat is a shrub grown in East Africa and southern Arabia, where some people chew its leaves for their mild stimulant effects. Synthetic cathinones are part of a group of drugs called new psychoactive substances (NPS). These are designed to mimic the effects of controlled substances and have no legitimate medical use. New substances are introduced into the market in quick succession to evade or hinder law enforcement efforts to address their manufacture and sale.

Not much is known yet about how synthetic cathinones affect the human brain especially neurotoxic mechanisms that lead to fatalities. In both humans and animals, the commonly encountered neuro-clinical toxic syndromes (also known as toxidromes) are sympathomimetic and hallucinogenic, increasing the risk of excited/agitated delirium syndrome and serotonin syndrome. Sympathomimetic toxidromes are characterized by neurological and psychiatric symptoms, such as agitation, anxiety, delusions, hyperactivity, paranoia and seizures, alongside diaphoresis, hyperthermia, and mydriasis. The hallucinogenic toxidrome includes disorientation, hallucinations, memory disruption, psychotic episodes [125,126].

The observed concentrations of synthetic cathinones vary widely and often have overlapping ranges for fatal and nonfatal cases. To draw conclusions about the cause of death it is necessary to know the drug concentrations and also the clinical situation of each case [116,127]. Furthermore, currently, there are difficulties related to the few studies that specifically explain the neurotoxic mechanism underlying death.

Based upon the presented data, it is not possible to establish the exact lethal concentrations of cathinones. Where there is clinical suspicion of death due to synthetic cathinones intoxication, postmortem blood concentrations >1 lg/mL (in persons without developed tolerance) can be cautiously considered as lethal concentrations. This compilation of cathinones concentrations can be a guide to find papers on a specific compound; however, they cannot be taken as absolute values to determine the cause of death.

In conclusion, in the light of the increasing prevalence of these drugs and the continued introduction of new molecules to the drug market, studies on the neurotoxicity mechanism caused by SCs are still inadequate. Despite many studies in recent years, there is still a great deal to clarify about the action of these substances of abuse on the brain.

## Figures and Tables

**Figure 1 ijms-24-06230-f001:**
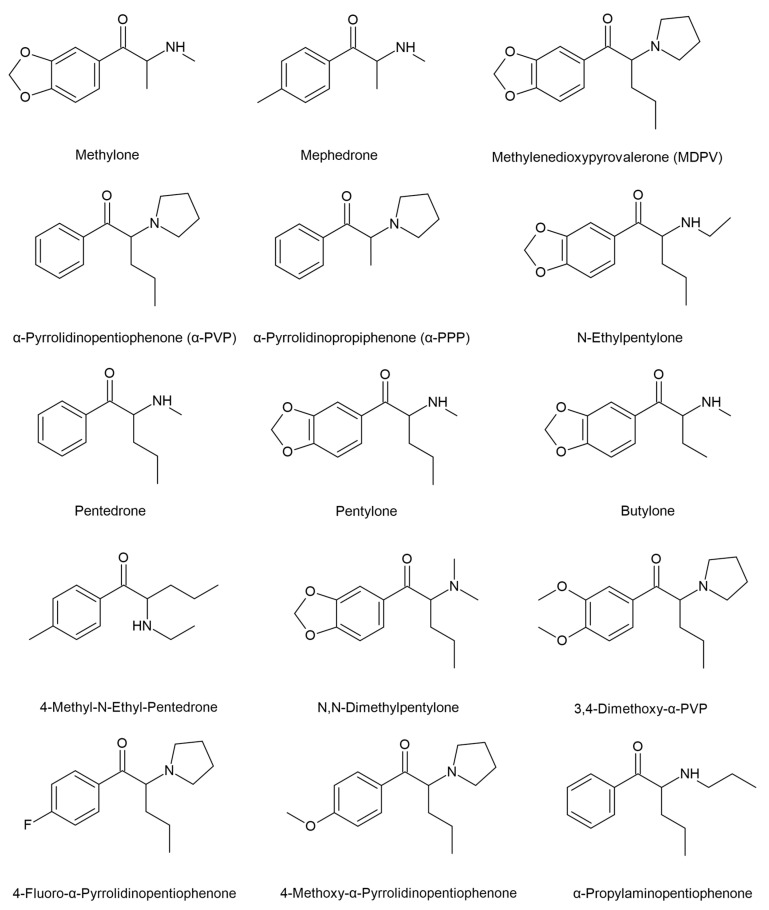
SCs used as drugs of abuse with common names and chemical structures.

**Figure 2 ijms-24-06230-f002:**
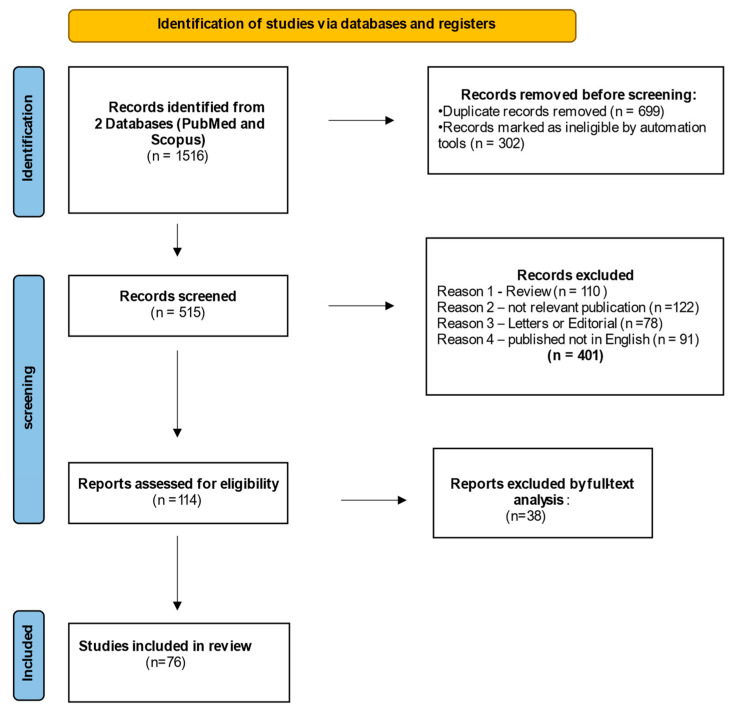
PRISMA flow diagram summarizing the literature selection process.

**Figure 3 ijms-24-06230-f003:**
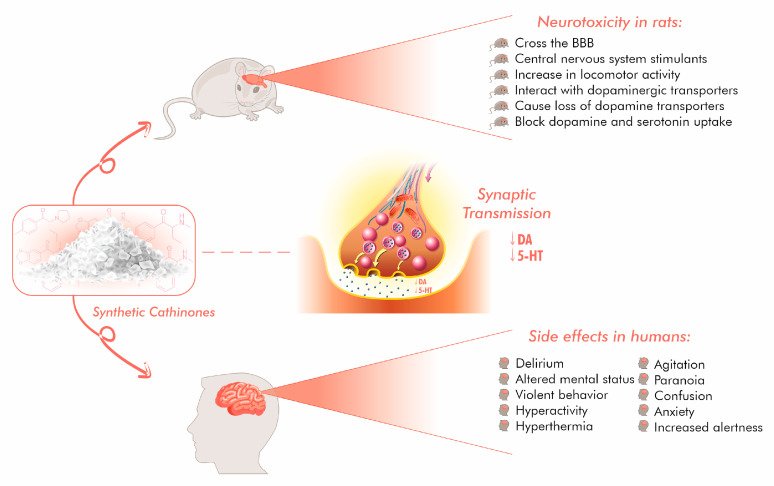
SCs’ mechanism of action and effects on rat and human brain. Synthetic cathinones act as blockers or substrates of DAT, SERT and NET. In this way they increase DA (dopamine), 5-HT (serotonon) or NA (norepinephrine) neurotransmission.

**Figure 4 ijms-24-06230-f004:**
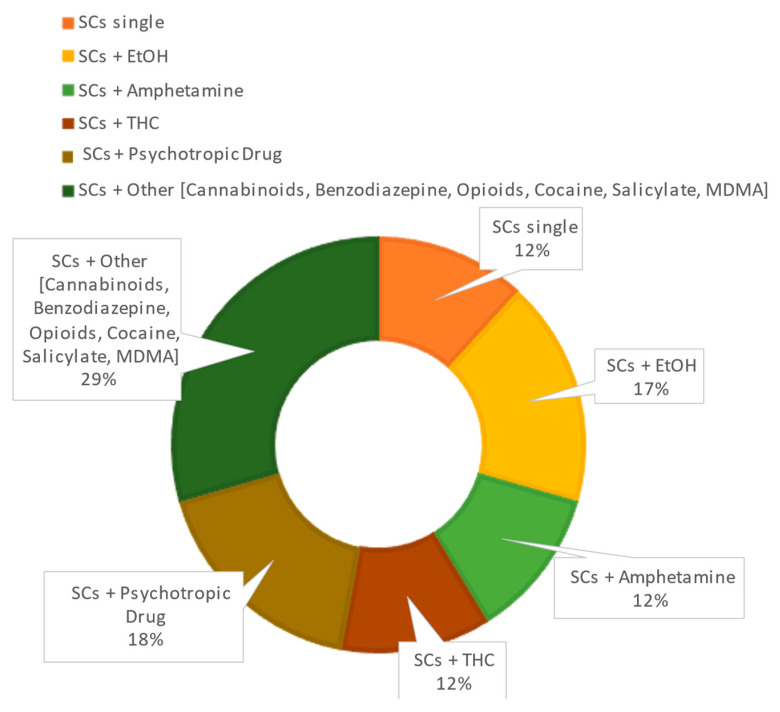
Distribution of other drugs of abuse in SC-related fatalities.

**Figure 5 ijms-24-06230-f005:**
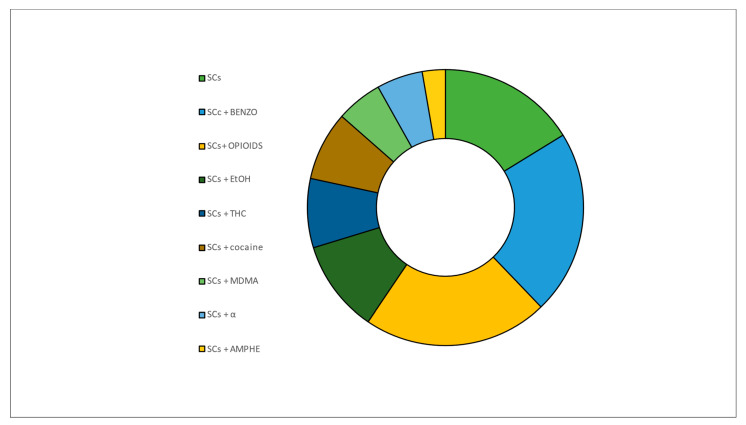
Association between SCs and other drugs in intoxication cases.

**Figure 6 ijms-24-06230-f006:**
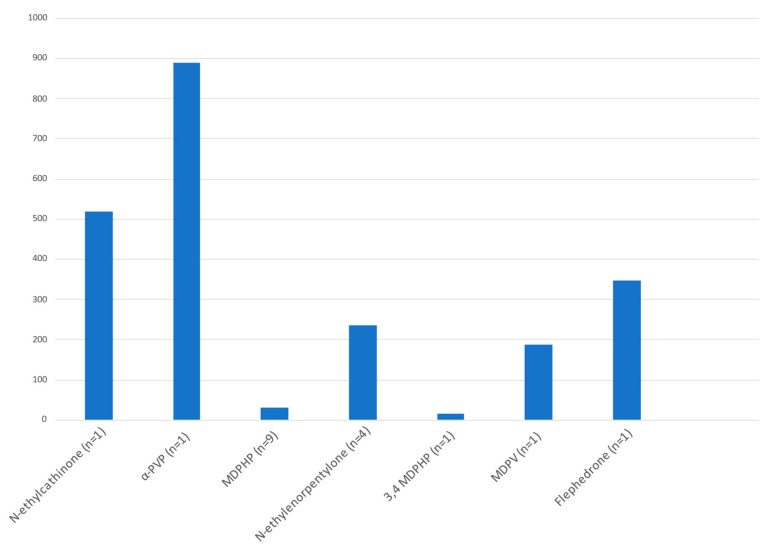
Different blood concentrations of SCs in intoxication cases.

## Data Availability

Not applicable.

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
