# Peer review of "Synthetic Cathinones and Neurotoxicity Risks: A Systematic Review"

_ijms, 2023, doi:10.3390/ijms24076230_

Round 1

Reviewer 1 Report (Previous Reviewer 1)

The Authors substancialy improved their manuscript.

Reviewer 2 Report (Previous Reviewer 2)

The author and colleagues have answered and revised all the questions raised. Therefore, I support the publication of this study.

Reviewer 3 Report (Previous Reviewer 4)

Manuscript was corrected.

This manuscript is a resubmission of an earlier submission. The following is a list of the peer review reports and author responses from that submission.

Round 1

Reviewer 1 Report

In this paper the Authors review the neurotoxic properties of synthetic cathinones. These new substances of abuse are derivatives from the alkaloid found in fresh khat leaves that are used by some Arabic cultures in social gatherings or to reduce fatigue and hunger. Unfortunately, their use as recreation drugs is becoming an increasing problem in the considered “developed” countries. Their toxic properties on the nervous system seem to be multiple but yet understudied, since the number of relevant publications is still low. This is a very well written review, covering most of the available bibliography.

Major concern:

The paper is well written and reviews an actual issue of growing concern. However, another similar and more extensive review has been published, overlapping some of the findings og this manuscript and weakening its relevance. (Archives of Toxicology (2021) 95:2895–2940 https://doi.org/10.1007/s00204-021-03083-3)

Minor suggestions:

Introduction

Line 67: consumption instead of assumption

Materials and Methods:

In Figure 2 the Authors show a n=32 for the studies included in the reviewed after the screening process. However, they included 62 references in the References section. What originates this discrepancy?

Reviewer 2 Report

Gloria Daziani and colleagues provides an overview of synthetic cathinones and neurotoxicity risks, and the current research performance and knowledge structure on this topic were performed by a systematic literature search.

It is an interesting work research, but there are still some things that need to be revised.

Overall, I would support the publication of this study after minor revision.

Comments:

1. The abstract should be factual and should contain background, results and conclusions.

2. List the most important future perspectives.

3. Can you give a strong reason why this work is new and what it will bring to the scientific community.

4. Check the punctuation of the full text, line 42 is missing a full stop.

5. Line 73: collecting different studies on animal models and humans.. There are many kinds of animal models. Why did the author mainly introduce rat.

6. Line 84: Why did you choose this time frame?

Reviewer 3 Report

Synthetic cathinones constitute a second largest and popular group of new psychoactive substances, so not surprisingly several excellent reviews on their actions, including neurotoxicity have been published.  

The major drawback of this review is a very limited number of cited articles. In my opinion at least the following papers are missing:

Human studies:

Adamowicz P. Clin Toxicol (Phila). 2021

Adamowicz P, Jurczyk A, Gil D, Szustowski S. Leg Med (Tokyo). 2020

Pieprzyca E, Skowronek R, Czekaj P. J Anal Toxicol. 2022

Domagalska E, Banaszkiewicz L, Woźniak MK, Kata M, Szpiech B, Kaliszan M. J Anal Toxicol. 2021

Beck O, Franzén L, Bäckberg M, Signell P, Helander A. Clin Toxicol (Phila). 2016

Bäckberg M, Lindeman E, Beck O, Helander A. Clin Toxicol (Phila). 2015

Beck O, Bäckberg M, Signell P, Helander A. Clin Toxicol (Phila). 2018

Pieprzyca E, Skowronek R, Korczyńska M, Kulikowska J, Chowaniec M. Leg Med (Tokyo). 2018

Adamowicz P, Hydzik P. Clin Toxicol (Phila). 2019

Grapp M, Kaufmann C, Schwelm HM, Neukamm MA, Blaschke S, Eidizadeh A. Drug Test Anal. 2020

Nakamura M, Takaso M, Takeda A, Hitosugi M. Leg Med (Tokyo). 2022

Thornton SL, Gerona RR, Tomaszewski CA. J Med Toxicol. 2012

Costa JL, Cunha KF, Lanaro R, Cunha RL, Walther D, Baumann MH. Drug Test Anal. 2019

Animals studies:

Espinosa-Velasco M, Reguilón MD, Bellot M, Nadal-Gratacós N, Berzosa X, Gómez-Canela C, Rodríguez-Arias M, Camarasa J, Escubedo E, Pubill D, López-Arnau R. Prog Neuropsychopharmacol Biol Psychiatry. 2022

Gatch MB, Shetty RA, Sumien N, Forster MJ. Addict Biol. 2021

Gatch MB, Rutledge MA, Forster MJ. Psychopharmacology (Berl). 2015

Gatch MB, Dolan SB, Forster MJ. Drug Alcohol Depend. 2019

Marusich JA, Grant KR, Blough BE, Wiley JL. Neurotoxicology. 2012

Manke HN, Nelson KH, Vlachos A, Bailey JM, Maradiaga KJ, Weiss TD, Rice KC, Riley AL. Neurotoxicol Teratol. 2021

Li J, Lin Z, Tao X, Huang Z, Zhang Y, Zheng S, Wang H, Rao Y. Behav Pharmacol. 2019

Duart-Castells L, Nadal-Gratacós N, Muralter M, Puster B, Berzosa X, Estrada-Tejedor R, Niello M, Bhat S, Pubill D, Camarasa J, Sitte HH, Escubedo E, López-Arnau R. Neuropharmacology. 2021

Gannon BM, Baumann MH, Walther D, Jimenez-Morigosa C, Sulima A, Rice KC, Collins GT. Neuropsychopharmacology. 2018

Schiavi S, Melancia F, Carbone E, Buzzelli V, Manduca A, Peinado PJ, Zwergel C, Mai A, Campolongo P, Vanderschuren LJMJ, Trezza V. Neuropsychopharmacology. 2020

Wojcieszak J, Kuczyńska K, Zawilska JB. Neurotox Res. 2021

Lopez-Arnau R, Duart-Castells L, Aster B, Camarasa J, Escubedo E, Pubill D. Psychopharmacology (Berl). 2019

Schindler CW, Thorndike EB, Walters HM, Walther D, Rice KC, Baumann MH. Addict Biol. 2020

Gregg RA, Baumann MH, Partilla JS, Bonano JS, Vouga A, Tallarida CS, Velvadapu V, Smith GR, Peet MM, Reitz AB, Negus SS, Rawls SM. Br J Pharmacol. 2015

Fantegrossi WE, Gannon BM, Zimmerman SM, Rice KC. Neuropsychopharmacology. 2013

Suyama JA, Sakloth F, Kolanos R, Glennon RA, Lazenka MF, Negus SS, Banks ML. J Pharmacol Exp Ther. 2016

Kolesnikova TO, Khatsko SL, Eltsov OS, Shevyrin VA, Kalueff AV. Neurotoxicol Teratol. 2019

Wojcieszak J, Andrzejczak D, Wojtas A, Gołembiowska K, Zawilska JB. Forensic Toxicol. 2018

Botanas CJ, Yoon SS, de la Peña JB, Dela Peña IJ, Kim M, Woo T, Seo JW, Jang CG, Park KT, Lee YH, Lee YS, Kim HJ, Cheong JH. Pharmacol Biochem Behav. 2017

Cheong JH, Choi MJ, Jang CG, Lee YS, Lee S, Kim HJ, Seo JW, Yoon SS. Psychopharmacology (Berl). 2017

Atehortua-Martinez LA, Masniere C, Campolongo P, Chasseigneaux S, Callebert J, Zwergel C, Mai A, Laplanche JL, Chen H, Etheve-Quelquejeu M, Mégarbane B, Benturquia N. J Psychopharmacol. 2019

Schindler CW, Thorndike EB, Goldberg SR, Lehner KR, Cozzi NV, Brandt SD, Baumann MH. Psychopharmacology (Berl). 2016

Berquist MD 2nd, Traxler HK, Mahler AM, Baker LE. Drug Alcohol Depend. 2016

Nadal-Gratacós N, Lleixà E, Gibert-Serramià M, Estrada-Tejedor R, Berzosa X, Batllori X, Pubill D, Camarasa J, Escubedo E, López-Arnau R. Int J Mol Sci. 2022

Glatfelter GC, Walther D, Evans-Brown M, Baumann MH. ACS Chem Neurosci. 2021

Allen SA, Tran LH, Oakes HV, Brown RW, Pond BB. Neurotox Res. 2019

Sewalia K, Watterson LR, Hryciw A, Belloc A, Ortiz JB, Olive MF. Neuropharmacology. 2018

Saha K, Li Y, Holy M, Lehner KR, Bukhari MO, Partilla JS, Sandtner W, Sitte HH, Baumann MH. Psychopharmacology (Berl). 2019

Saha K, Partilla JS, Lehner KR, Seddik A, Stockner T, Holy M, Sandtner W, Ecker GF, Sitte HH, Baumann MH. Neuropsychopharmacology. 2015

Adám A, Gerecsei LI, Lepesi N, Csillag A. Neurotoxicology. 2014

Xu P, Qiu Y, Zhang Y, Βai Y, Xu P, Liu Y, Kim JH, Shen HW. Int J Neuropsychopharmacol. 2016

Marusich JA, Gay EA, Watson SL, Blough BE. Eur J Pharmacol. 2021

Bernstein DL, Nayak SU, Oliver CF, Rawls SM, Rom S. Neurosci Res. 2020

Wojcieszak J, Andrzejczak D, Wojtas A, Gołembiowska K, Zawilska JB. Neurotox Res. 2019

Colon-Perez LM, Tran K, Thompson K, Pace MC, Blum K, Goldberger BA, Gold MS, Bruijnzeel AW, Setlow B, Febo M. Neuropsychopharmacology. 2016

Cameron K, Kolanos R, Vekariya R, De Felice L, Glennon RA. Psychopharmacology (Berl). 2013

Cytotoxicity:

Wojcieszak J, Andrzejczak D, Woldan-Tambor A, Zawilska JB. Neurotox Res. 2016

Other comments:

1.     Authors, please provide full names for all abbreviations used in the text.

2.     Lines 24-25. Tachycardia and hyponatremia are not neurological symptoms.

3.     Line 27. “the ability to modulate serotonin”???

4.     Lines 58 and 59. “Regarding their mechanism of action, SCs can inhibit serotonin (5-HT) reuptake 58 transporters and dopamine (DA) and norepinephrine reuptake transporters”. This statement should be corrected. “Mechanisms of action for synthetic cathinones involve interactions with dopamine, serotonin and norepinephrine transporters with varying affinities and selectivities, as shown in in vitro studies, including human cell lines and preclinical modelsRing-substituted cathinones, such as methylone, act as transporter substrates that increase the release of dopamine, serotonin and norepinephrine. The presence of a pyrrolidine ring, as in alpha-PVP, act as transport blockers (reuptake inhibitors) at the dopamine transporter (. Increasing the length of the alpha-carbon chain increased affinity and potency at the dopamine transporter. Compounds with a higher potency at the dopamine transporter, including alpha-pyrrolidinophenones and 4-fluoroamphetamine (4-FA), exhibit stimulant properties similar to methamphetamine while cathinones that have similar potencies at dopamine and serotonin transporters, or higher potency at the serotonin transporter may have more empathogenic activity (i.e., ethylone)” (cited from Logan et al., 2017).

5.     Figure 1 presents chemical structures of 21 synthetic cathinones out of more than 160 identified. What was a criterion to choose these particular ones?

6.     Figure 3. Synthetic cathinones act as blockers or substrates od DAT, SERT and NET. In this way they increase DA, 5-HT or NA neurotransmission.

7.     Line 239. “typical of “ ???

8.     Line 245. PEA arrest, should be pulseless electrical activity

9.      Last but not least. The manuscript should corrected by a native English speaker.

Reviewer 4 Report

Synthetic cathinones are part of a group of drugs that concern public health officials called new psychoactive substances (NPS). NPS are unregulated psychoactive mind-altering substances with no legitimate medical use and are made to copy the effects of controlled substances. They are introduced and reintroduced into the market in quick succession to dodge or hinder law enforcement efforts to address their manufacture and sale. Synthetic cathinones are marketed as cheap substitutes for other stimulants such as amphetamines and cocaine. Much is still unknown about how synthetic cathinones affect the human brain. Therefore, the theme of presenteted review is timely and is important for a public health.

However, in my opinion manuscript requires corrections. The topic presented by the Authors is very interesting, however, it was presented in too general way. After such publications (reviews), we expect much more information and a more accurate presentation of the mechanism of suggested changes. The Authors should unify the information’s on the studies contained in the tables, organize them either by the used substance or, for example, by the year of publication. In addition, the mechanisms responsible for the neurotoxic effects of the substances should be examined in more detail, also taking into account their adverse impact on the memory and development of other addictions, as well as different age groups (the age at which we use addictive substances is of key importance). It is also worth presenting more recent publications.